# Self-Supervised Learning of Brain Dynamics from Broad Neuroimaging Data

**Armin W. Thomas**
Department of Psychology
Stanford University
athms@stanford.edu

**Christopher Ré**
Department of Computer Science
Stanford University
chrismre@stanford.edu

**Russell A. Poldrack**
Department of Psychology
Stanford University
poldrack@stanford.edu

## Abstract

Self-supervised learning techniques are celebrating immense success in natural language processing (NLP) by enabling models to learn from broad language data at unprecedented scales. Here, we aim to leverage the success of these techniques for mental state decoding, where researchers aim to identify specific mental states (e.g., the experience of anger or joy) from brain activity. To this end, we devise a set of novel self-supervised learning frameworks for neuroimaging data inspired by prominent learning frameworks in NLP. At their core, these frameworks learn the dynamics of brain activity by modeling sequences of activity akin to how sequences of text are modeled in NLP. We evaluate the frameworks by pre-training models on a broad neuroimaging dataset spanning functional Magnetic Resonance Imaging data from $11,980$ experimental runs of $1,726$ individuals across $34$ datasets, and subsequently adapting the pre-trained models to benchmark mental state decoding datasets. The pre-trained models transfer well, generally outperforming baseline models trained from scratch, while models trained in a learning framework based on causal language modeling clearly outperform the others.

## 1   Introduction

In mental state decoding, researchers aim to identify certain mental states (e.g., deciding to accept or reject a gamble or the experience of happiness or fear) from brain activity [1]. To this end, researchers train predictive models to correctly identify (i.e., decode) these mental states from measured brain activity. At first sight, this approach seems straightforward, yet researchers interested in training mental state decoding models are faced with the challenge that individual neuroimaging datasets, particularly functional Magnetic Resonance Imaging (fMRI) data, are often of high dimension and low sample size, such that individual samples can comprise many hundred thousand dimensions while individual datasets include only a few hundred samples for each of tens to hundreds of individuals. In this setting, where the dimensionality of the data far exceed the number of samples, predictive models are prone to overfitting, which severely limits any generalizable insights that can be gained from training these models about the studied mental states and brain activity.

In spite of the low sample size of individual datasets, neuroimaging research has recently entered a big data era, as individual researchers more frequently share their collected datasets publicly [2, 3]. In addition, several efforts have been made to standardize the data structure [4] and preprocessing [5] of neuroimaging data. Together, these developments open up new possibilities for the application of pre-training in neuroimaging at scale, by allowing for the pre-training of models on broad, public neuroimaging data so that the resulting models can be adapted well to the datasets collected by individual researchers, without the need for much new data.

A wealth of empirical evidence has demonstrated that models pre-trained on large neuroimaging datasets achieve better mental state decoding performances on new data, while also requiring less

36th Conference on Neural Information Processing Systems (NeurIPS 2022).

training time and data, than models trained from scratch (e.g., [6, 7, 8, 9]). Yet, much of this work is limited by either pre-training models on large but homogeneous datasets, such as data from many individuals who all perform the same few tasks at the same few acquisition sites [10, 7] (jeopardising the generalizability of the resulting models due to potential systematic biases in homogeneous datasets, e.g., specific to the acquisition site or experimental paradigm [11, 12, 13, 14, 15]), or by requiring highly-preprocessed input data (e.g., statistical maps summarizing the measured sequences of brain activity [16]). In addition, researchers often use standard supervised pre-training tasks by tasking models with identifying the specific mental state assigned to each sample of the data [6, 7, 10, 17]. While this kind of supervised training can be fruitful within individual neuroimaging datasets, it is difficult to extend to many datasets, as neuroimaging researchers generally do not adhere to standardized labeling schemes for mental states [18] when assigning labels to the mental states of their experiments. Due to this lack of standardization, it is often unclear whether two datasets contain the same or distinct mental states (for a detailed discussion, see [19]).

Here, we aim to overcome these limitations by leveraging recent advances in self-supervised learning to pre-train deep learning (DL) models on broad neuroimaging data. In particular, we take inspiration from natural language processing (NLP), where the application of self-supervised learning has led to unprecedented breakthroughs in the adaptive capabilities of pre-trained models (e.g., [20, 21, 22, 23]), and test whether modeling sequences of brain activity akin to sequences of text enables the learning of generalizable representations of brain activity. To enable the application of NLP modeling techniques to neuroimaging data, we represent sequences of brain activity akin to the embedded representation of words in sequences of text in NLP [24, 25] by representing each measurement time point as a vector that describes whole-brain activity as the activity of a set of functionally-independent brain networks (s.t. each vector value indicates the activity of one of the networks at this point in time [26]).

**Contributions.** By representing sequences of brain activity similar to the embedded representation of text in NLP, we are able to devise novel self-supervised learning frameworks for neuroimaging data that are inspired by prominent learning frameworks in NLP, namely, sequence-to-sequence autoencoding [27], causal language modeling [20], and masked language modeling [21]. We pre-train DL models with these newly devised frameworks on a large-scale neuroimaging dataset comprising fMRI data from $11,980$ experimental runs of $1,726$ individuals across $34$ datasets. To our knowledge, this represents one of the broadest neuroimaging datasets used for pre-training to date, spanning a wealth of experimental conditions and a diverse set of acquisition sites. We evaluate the downstream adaptation performance of the pre-trained models in two benchmark mental state decoding datasets and show that models pre-trained in a learning framework based on causal language modeling clearly outperform the others, while all pre-trained models generally outperform baseline models trained from scratch. To enable others to build on this work, we make our code, training data, and pre-trained models publicly available [1].

## 2 Methods overview

### 2.1 Input data: from brain volumes to brain networks

FMRI data are conventionally represented in four dimensions, such that the measured blood-oxygen-level-dependent (BOLD) signal is indicated in temporal sequences $S = \{V_1, ..., V_t\}$ of 3-dimensional volumes $V \in \mathbb{R}^{x \times y \times z}$ that show the BOLD signal intensity for each spatial location of the brain (as indicated by the three spatial dimensions $x$, $y$, and $z$). However, due to the strong spatial correlations of brain activity, the measured BOLD signal is often summarized differently by representing it as a set $\Theta = \{\theta_1, ..., \theta_n\}$ of $n$ brain networks (i.e., parcels) $\theta$, each describing the BOLD signal for some subset of voxels $v_{x,y,z} \subset V$ with correlated activity patterns.

Here, we utilize such a functional parcellation of brain activity to represent the measured BOLD signal, namely, Dictionaries of Functional Modes (DiFuMo), which was recently proposed by [28] and learned across millions of fMRI volumes. DiFuMo defines $\Theta$ through a sparse dictionary matrix $D \in \mathbb{R}^{p \times n}$ containing $n$ $p$-dimensional networks (where each dimension indicates a voxel $v \in V$, flattened over the three spatial dimensions). The BOLD signal of volume $V_t$ is then described as a linear combination of these networks by the use of weights $\alpha_t \in \mathbb{R}^n$, such that $V_t = D\alpha_t$ and $\alpha_t = D^\dagger V_t$, where $D^\dagger$ indicates the pseudo-inverse of $D$: $D^\dagger = (D^T D)^{-1} D^T \in \mathbb{R}^{n \times p}$. In this

---

[1] github.com/athms/learning-from-brains

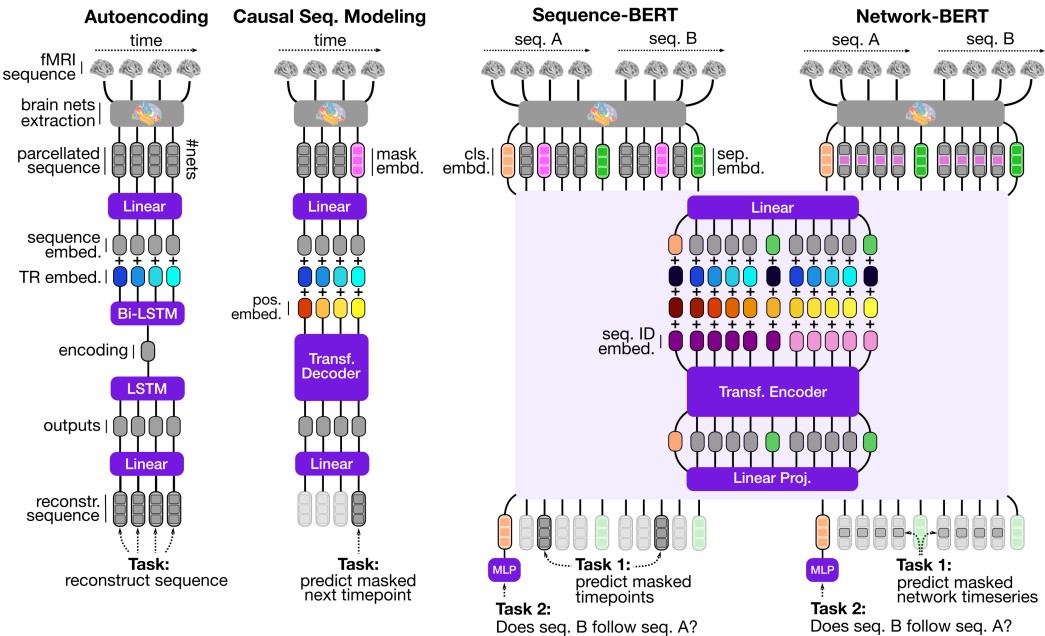

Figure 1: Proposed self-supervised learning frameworks for neuroimaging data, inspired by NLP.

work, we utilize the DiFuMo parcellation with $n = 1024$ networks, such that the resulting parcellated BOLD data $X \in \mathbb{R}^{t \times n}$ describe the measured BOLD signal of each of these networks for each time point $t$ of the input sequence $S$. This representation of the input data is analogous to the embedded representation of sentences in NLP, where each word is represented by some embedding vector [24, 25].

## 2.2 Self-supervised learning frameworks: modeling sequences of brain activity akin to sequences of text

We first establish several commonalities among the frameworks before describing them individually in more detail (for an overview, see Fig. 1).

### 2.2.1 Commonalities among frameworks

First, each framework takes as input a parcellated BOLD sequence $X \in \mathbb{R}^{t \times n}$ (with $t$ time points for $n$ networks) as well as the corresponding sampling frequency (i.e., repetition time or TR) at which the BOLD data were collected (typically in the range of $0.7$ s to $4$ s).

Second, all frameworks linearly project the parcellated BOLD sequence $X$ into an embedding representation $E^X \in \mathbb{R}^{t \times e}$: $E^X_{t,e} = b_e + \sum_n X_{t,n} w_{n,e}$, where $w \in \mathbb{R}^{n \times e}$ and $b \in \mathbb{R}^e$ indicate weights and biases, thereby reducing the dimensionality of the input: $\mathbb{R}^{t \times n} \to \mathbb{R}^{t \times e}$.

Third, to account for any differences in the sampling frequency of the BOLD sequences, the frameworks add one of $r$ learnable TR embeddings $E^{TR} \in \mathbb{R}^{r \times e}$ to each time point $t \in E^X$. Each TR embedding corresponds to one possible time point $k$ of the input. We set $r = 1,500$ with a time increment of $0.2$ s between subsequent TR embeddings, such that $k \in 0, 0.2, ..., 300$ s. For example, for an input sequence of $21$ s that was collected at a TR of $0.7$ s, we first round each time point $t \in 0, 0.7, ..., 21$ to the closest $k$ and then add the corresponding $E^{TR}_k$ to $E^X_k$. Note that we always label time points relative to the beginning of an input sequence, such that time point $k = 0$ is assigned to the first sequence element. Importantly, TR embeddings carry different information than the position embeddings commonly used in transformer models [29] because the same position in an input can correspond to different time points, depending on the sampling frequency at which the data were collected.

Fourth, all frameworks utilize standard sequence-to-sequence DL model architectures $f(\cdot)$ that take some embedded sequence $E^{in} \in \mathbb{R}^{t \times e}$ as input, e.g., defined as the sum of input and TR embeddings: $E^{in} = \{E_t^X + E_t^{TR}\}^{t \in E^X}$, and predict a corresponding output sequence $f(E^{in}) \in \mathbb{R}^{t \times e}$.

Fifth, during upstream learning, the output sequence $f(E^{in})$ is linearly projected back to the dimensionality of the parcellated BOLD sequence $X$, such that $\hat{X}_{t,n} = b_n + \sum_e f(E^{in})_{t,e} w_{e,n}$, where $w$ and $b$ again indicate weights and biases. This step is crucial, as all proposed learning frameworks involve reconstructing $X$ (or parts of $X$) after some of its elements have been masked or it has been encoded in a lower-dimensional representation. To measure reconstruction performance, we utilize the mean absolute difference between the parcellated BOLD sequence $X$ and its reconstruction $\hat{X}$: $L_{rec} = \frac{1}{n(J)} \sum_{j \in J} |X_j - \hat{X}_j|$, where $J$ indicates the set of elements that is to be reconstructed and $n(J)$ the number of elements in $J$.

Sixth, for downstream adaptation to a new mental state decoding task, the frameworks utilize a simple decoding head $p(\cdot)$, composed of a dense hidden layer with $e$ model units (one for each embedding dimension, with $tanh$ activation) as well as a $softmax$ output layer with one model unit $i$ for each considered mental state in the data. $p(x)_i$ therefore indicates the probability that input $x$ belongs to mental state $i$. Consequently, all frameworks treat the decoding task as a multinomial classification problem and use a standard cross entropy loss objective: $L_{cls} = - \sum_i y_i \log p(x)_i$, where $y_i$ indicates a binary variable that equals 1 if $i$ is the correct mental state and 0 otherwise.

### 2.2.2 Autoencoding

The autoencoding framework is inspired by recurrent neural machine translation models (e.g., [27]). These models take as input some text sequence, encode this sequence in a lower-dimensional representation, and predict an output sequence from the lower-dimensional encoding (e.g., a translation to another language).

Accordingly, the autoencoding framework uses a standard recurrent autoencoder architecture $f_a(\cdot)$, with an encoder and decoder part, each comprised of a stack of long short-term memory (LSTM [30]) units. The encoder $e(\cdot)$ takes $E^{in}$ as input and encodes it in a lower-dimensional representation $h \in \mathbb{R}^e$. The decoder $d(\cdot)$ then takes $h$ as input and predicts a corresponding output sequence $d(h) \in \mathbb{R}^{t \times e}$, such that $f_a(E^{in}) = d(h)$. Our encoder is composed of a stack of bidirectional LSTM units, whereas the decoder is build from a stack of unidirectional LSTM units. Note that we apply residual connections between the LSTM units in a stack: Let $LSTM^i$ and $LSTM^{(i+1)}$ be the $i$-th and $(i+1)$-th LSTM units in a stack (with weights $w^i$ and $w^{i+1}$). At the $t$-th step of an input sequence, the inputs and outputs of units $i$ and $(i+1)$ are defined as: $h_t^i, c_t^i = LSTM^i(h_{t-1}^i, c_{t-1}^i, x_{t-1}^{i-1}; w^i)$; $x_t^i = h_t^i + x_{t-1}^{i-1}$; $h_{t+1}^{i+1}, c_{t+1}^{i+1} = LSTM^{i+1}(h_{t+1}^{i+1}, c_{t+1}^{i+1}, x_t^i; w^{i+1})$, where $x_t^i \in \mathbb{R}^e$ represents the input to $LSTM^i$ at step $t$, whereas $h_t^i \in \mathbb{R}^e$ and $c_t^i \in \mathbb{R}^e$ represent hidden and cell state respectively. We define the sequence encoding $h$ that is forwarded to the decoder as the mean of the final hidden states of the two LSTM units contained in the encoder's final bidirectional LSTM unit.

**Upstream learning.** During pre-training, we jointly train the encoder and decoder to reconstruct the parcellated BOLD sequence $X$ by minimizing its mean absolute difference to the reconstructed sequence $\hat{X}$ (see section 2.2.1): $\min \frac{1}{t} \sum_{i=1}^t \frac{1}{n} \sum_{j=1}^n |\hat{X}_{i,j} - X_{i,j}|$. We also apply teacher-forcing [31] by using $E^{in}$ in 50% of the cases as each next input for the decoder instead of its own preceding output.

**Adaptation.** To adapt the autoencoding framework to a new decoding task, we forward $h$ as input to a corresponding decoding head $p(\cdot)$ (see section 2.2.1), thereby omitting the decoder $d(\cdot)$.

### 2.2.3 Causal sequence modeling (CSM)

The causal sequence modeling (CSM) framework is inspired by recent advances in causal language modeling [20], where models are trained to predict the next word in a text sequence. Specifically, these models receive some text sequence as input, represented as a matrix $X \in \mathbb{R}^{n \times e}$ with $n$ words each encoded by an $e$-dimensional vector, where the last sequence element (representing the last word) is masked, for example, by replacing it with a learned mask embedding $E^{msk} \in \mathbb{R}^e$ (e.g., "When it rains, it [msk]"). The model's task is then to predict the masked word. In causal language

modeling, models are thereby solely concerned with the part of the input sequence that precedes the masked word.

To adapt causal language modeling to neuroimaging data, we first mask the last time point $T$ of a parcellated BOLD sequence $X$ by replacing it with a learnable mask embedding $E^{msk}$ before transforming $X$ to $E^{in}$ by means of linear projection and the addition of TR embeddings $E^{TR}$. In line with recent advances in causal language modeling, we utilize a standard transformer decoder model $f_d(\cdot)$ (based on the GPT architecture [20]) in this learning framework [2]. As transformer models are not aware of the ordering of their input, we further add learnable position embeddings $E^{pos} \in \mathbb{R}^{p \times e}$ to $E^{in}$, each representing one of $p$ possible positions in the input sequence (we set $p = 512$ for all transformer models). Given $E^{in}$, the transformer decoder model predicts a corresponding output sequence $f_d(E^{in}) \in \mathbb{R}^{t \times e}$ in the form of the hidden states at the output of its last layer.

**Upstream learning.** In line with causal language modeling, the upstream objective of the CSM framework is to reconstruct the masked last time point $T$ of the parcellated BOLD sequence $X$ by minimizing the mean absolute difference to its reconstruction $\hat{X}_T$: $\min \frac{1}{n} \sum_{i=1}^{n} |\hat{X}_{T,i} - X_{T,i}|$.

**Adaptation.** During downstream adaptation, we attach a learnable classification embedding $E^{cls} \in \mathbb{R}^n$ to the end of BOLD parcellation $X$ and forward the resulting prediction $f(E^{in})_{T+1}$ to a corresponding decoding head $p(\cdot)$ (see section 2.2.1).

### 2.2.4 Sequence-BERT

The Sequence-BERT framework is based on recent advances in bidirectional masked language modeling, particularly BERT [21]. BERT is trained to jointly solve a masked-language-modeling and next-sentence-prediction task. To this end, BERT receives two masked sentences $S^a$ (e.g., "The sky is [msk].") and $S^b$ (e.g., "The [msk] is shining.") as input, in which some fraction of words have been randomly masked. BERT is then asked to i) predict the masked words (masked-language-modeling) and ii) determine whether $S^b$ follows $S^a$ (next-sentence-prediction). In contrast to causal language modeling, masked language modeling is considered as a bidirectional learning task, as the model is not just concerned with the parts of the input that precede a masked word but also the parts that follow it.

To adapt BERT to neuroimaging data, we first sample two input sequences $X^a \in \mathbb{R}^{t^a \times n}$ and $X^b \in \mathbb{R}^{t^b \times n}$ from the data (with lengths $t^a$ and $t^b$). In $50\%$ of the cases, the two sequences represent sequential parts of the same underlying BOLD sequence, while in the other $50\%$ of cases, the two sequences are randomly sampled from two distinct BOLD sequences. Due to the strong temporal autocorrelation of BOLD data, we randomly leave a gap of 1-5 TRs (generally corresponding to a gap of 1 to 20 s) between the two sequences when sampling them from the same underlying BOLD sequence (to encourage the models to consider the entire input sequences and not just their connecting time points). In line with BERT, we next randomly mask time points of each sequence (with a probability of $20\%$) by replacing them with a learnable mask embedding $E^{msk}$. We also attach a separation embedding $E^{sep} \in \mathbb{R}^n$ to the end of each sequence (to indicate their endpoints) and a classification embedding $E^{cls}$ to the beginning of $S^a$ (for the next-sentence-prediction task; see upstream learning below) [3]. The resulting versions of $X^a$ and $X^b$ are then linearly projected to obtain respective BOLD embeddings $E^{X^a}$ and $E^{X^b}$ to which we further add TR and position embeddings $E^{TR}$ and $E^{pos}$ as well as sequence ID embeddings $E^{id} \in \mathbb{R}^{2 \times e}$ to indicate each of the two input sequences. The resulting embedding representation $E^{in} = [\{E_t^{X^a} + E_t^{TR} + E_t^{pos} + E_1^{id}\}^{t \in t^a}, \{E_t^{X^b} + E_t^{TR} + E_t^{pos} + E_2^{id}\}^{t \in t^b}]$ is then forwarded to a standard transformer encoder model $f_e(\cdot)$ (based on the BERT architecture [21]), which outputs a corresponding output sequence $f_e(E^{in}) \in \mathbb{R}^{t \times e}$ in the form of the hidden states at the output of its final layer.

**Upstream learning.** Similar to BERT, we define the upstream objective of the Sequence-BERT framework as the sum of the learning objectives of its two tasks: In line with masked-language-

---

[2]While the described input masking procedure is not needed for transformer decoder models, due to their causal attention mechanism, we chose this formulation of the task to ensure that the CSM framework also generalizes to other model architectures.

[3]We use the same learnable "dummy" TR embedding (see section 2.2.1 and Fig. 1) for any elements that are inserted into an input sequence during training (e.g., for classification or separation embeddings).

modeling, we define the first objective as the reconstruction loss $L_{rec}$ for all masked time points $J$: $L_{rec} = \frac{1}{n(J)} \sum_{j \in J} \frac{1}{n} \sum_{i=1}^{n} |X_{j,i} - \hat{X}_{j,i}|$. In line with next-sentence-prediction, we define the other objective as the binary cross entropy loss for a decoding head that receives $f(E^{in})_1$ (corresponding to the transformer output for classification embedding $E^{cls}$) as input and predicts probability $p$ that $X^b$ follows $X^a$: $L_{cls} = -y \log p - (1-y) \log(1-p)$, where $y$ indicates a binary variable that equals 1 if the two sequences come from the same fMRI run and 0 otherwise.

**Adaptation.**   During downstream adaptation, we forward the transformer's output for classification embedding $f_e(E^{in})_1$ as input to a corresponding decoding head $p(\cdot)$ (see section 2.2.1).

### 2.2.5   Network-BERT

All so far presented frameworks model their input on the level of the individual time points of a sequence by trying to reconstruct the parcellated whole-brain BOLD signal for specific time points. Yet, the dynamics of brain activity data can also be viewed differently by focusing on the interaction of network activities over time instead of on the distribution of whole-brain activity at individual time points. Consequently, we created a variant of the Sequence-BERT framework that randomly (at a rate of 10%) masks the activity time courses of individual networks $n$ in the parcellated BOLD sequence $X$ by replacing them with a learnable network mask embedding $E^{msk} \in \mathbb{R}^t$ (instead of masking whole-brain activity at individual time points). With the exception of the masking procedure, the Network-BERT and Sequence-BERT frameworks are identical.

**Upstream learning.**   During upstream learning, Network-BERT uses the same cross-entropy loss $L_{cls}$ as Sequence-BERT and adds it to a variant of the reconstruction loss that measures reconstruction performance for the masked network time courses $J$: $L_{rec} = \frac{1}{n(J)} \sum_{j \in J} \frac{1}{t} \sum_{i=1}^{t} |X_{i,j} - \hat{X}_{i,j}|$.

**Adaptation.**   During downstream adaptation, Network-BERT forwards the transformer's output for classification embedding $f_e(E^{in})_1$ to a respective decoding head $p(\cdot)$ (see section 2.2.1).

## 2.3   Training details

We train all models with stochastic gradient descent and the ADAM optimizer (with $\beta_1 = 0.9$, $\beta_2 = 0.999$, and $\epsilon = 1e^{-8}$) [32], if not reported otherwise. We also apply a linear learning rate decay schedule (with a warmup phase of 1% of the total number of training steps), gradient norm clipping at 1.0, and $L2$-regularisation (weighted by 0.1).

During upstream learning, we set the maximum learning rates to $2e^{-4}$, $5e^{-4}$, $1e^{-4}$, and $1e^{-4}$ for the autoencoding, CSM, Sequence-BERT, and Network-BERT frameworks, respectively (based on the learning rates used in corresponding NLP studies [20, 21, 27]), and randomly sample sequences $X$ of 10 to 55 TRs from our upstream fMRI runs. For the Sequence- and Network-BERT frameworks, each sequence is split into two sequences $X_i^a$ and $X_i^b$ of equal length (with a randomly sampled gap of 1 to 5 TRs between the two sequences) and $X_i^b$ randomly exchanged (with 50% probability) with another $X_j^b$ (where $i \neq j$).

During downstream adaptation, we begin training with the pre-trained model parameters and allow all parameters to be changed freely during training. Input sequences are drawn from the downstream fMRI runs according to the on- and offset defined for each experimental trial (with durations generally ranging between 2 and 30 s), when accounting for the temporal delay of the hemodynamic response function.

**Compute resources.**   Upstream training was performed on Google Compute Engine n1-highmem-64 nodes with four Nvidia Tesla P100 GPUs, 64 CPU threads, and 416 GB RAM memory, while downstream adaptations were performed on compute nodes of the Texas Advanced Computing Center with one Nvidia 1080-TI GPU, 32 CPU threads, and 128 GB RAM memory. Training times varied depending on model size but were generally in the range of 1 to 3 days for upstream learning and 0.5 to 3 hours for downstream adaptations.

## 3 Experiments

### 3.1 Datasets

All unprocessed fMRI data used in this study are publicly available through OpenNeuro.org [3] and the Human Connectome Project (HCP [33]). For an overview of our preprocessing, see Appendix A.3.

#### 3.1.1 Upstream: $11,980$ fMRI runs from $1,726$ individuals across $34$ datasets

Our upstream dataset comprises $11,980$ fMRI runs from $1,726$ individuals and $34$ datasets (see Fig. 2 and Appendix Table A1). To our knowledge, this represents one of the broadest fMRI datasets used for pre-training in neuroimaging to date, spanning many acquisition sites and a diverse set of experimental conditions and domains. A few prominent examples of included datasets are The Midnight Scan Club [34], Individual Brain Charting [35], Amsterdam Open MRI collection [36], BOLD5000 [37], and the Narratives collection [38]. We split the upstream data into distinct training and evaluation datasets by randomly designating $5\%$ of the fMRI runs of each included fMRI dataset as evaluation data (at a minimum of 2 runs per dataset) and using the rest of the runs for training. At each evaluation step, we randomly sample $640,000$ sequences from the evaluation dataset.

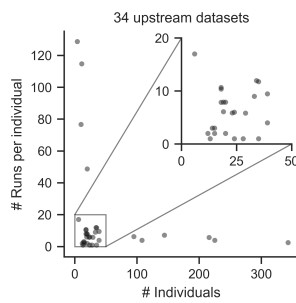

Figure 2: Number of individuals and average number of runs per individual of each upstream dataset.

#### 3.1.2 Downstream: benchmark datasets with many mental states

To evaluate the adaptation performance of the pre-trained models, we utilize two benchmark mental state decoding datasets that span a wide range of mental states, namely, task-fMRI data from the HCP [33] and the Multi-Domain Task Battery (MDTB [39]). Our HCP dataset spans 100 participants and 19 distinct mental states across seven experimental tasks (see Appendix A.2). For each task, two fMRI runs were collected. We also include two resting state fMRI runs per HCP participant, in which individuals simply rest in the fMRI without any particular task, increasing the total number of mental states to 20. The MDTB dataset spans 24 individuals and 26 experimental tasks, which participants performed across two scanning sessions with eight fMRI runs per session. Due to the strong similarity of the experimental conditions within individual experimental tasks of the MDTB, we define each of its tasks as one target mental state (see Appendix A.2).

### 3.2 Hyper-parameter evaluation: larger embeddings but not deeper models

In our first experiment, we compared the performance of different model hyper-parameter settings in each framework. Specifically, we evaluated three different embedding dimensions $e$ (192, 384, and 768; see section 2.2.1) and three different model depths (2, 4, and 6 hidden layers for the encoder and decoder parts of the autoencoding framework and 4, 8, and 12 hidden layers for the transformer models). In line with other work [40], we scaled the number of attention heads $n_\alpha$ per layer of the transformer models with the embedding dimension, such that $n_\alpha = \frac{e}{64}$. We trained one model for each point of the resulting 9-point grid of each framework for $200,000$ training steps, using a mini-batch size of 256 sequences. Overall, model performances improved (in terms of validation loss) with larger embedding dimensions but not with model depth (Fig. 3). We therefore decided to continue all further analyses with models comprising 768 embedding dimensions and 2 hidden layers for the encoder and decoder parts of the autoencoding framework and 4 hidden layers for the transformer models.

### 3.3 Upstream performance: models learn well in all frameworks

In our second experiment, we fully pre-trained one model in each framework according to our previously selected set of hyper-parameters (see section 3.2) by training models for $300,000$ training steps at a mini-batch size of 768 sequences. All models learned well over the course of their training,

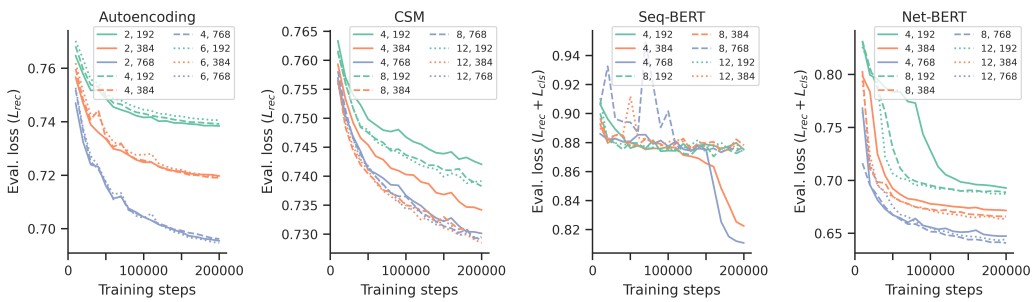

Figure 3: Hyper-parameter evaluation. Model performances improve with larger embedding dimensions but not with model depth. Note that we do not plot the evaluation loss for the Sequence-BERT variant with 12 hidden layers and 768 embedding dimensions because the loss of this model variant did not meaningfully change over the course of its training (see Appendix B.1).

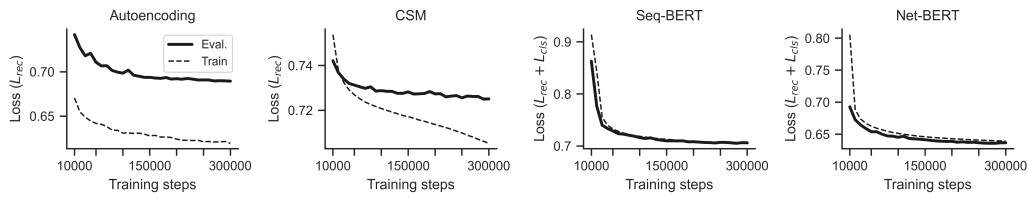

Figure 4: Upstream learning. Models learn well in each framework, with gradually decreasing training and evaluation losses.

with gradually decreasing training and validation losses (Fig. 4). Note that the transformer model trained in the CSM framework slightly overfitted the upstream data, whereas the gap in training and evaluation performance of the autoencoding framework results from the application of teacher-forcing during training (see section 2.2.2).

We also evaluated the brain activity reconstruction performance (as measured by $L_{rec}$; see section 2.2.1) of the final models for different parts of the brain and found that they exhibited similar overall distributions of reconstruction error throughout the brain, with relatively higher errors in the posterior parietal, occipital, and cingulate cortices as well parts of the limbic system (see Appendix B.2).

### 3.4 Adapting pre-trained language models does not yield upstream performance gains

Given that the formatting of our input data is highly similar to the embedded representation of sentences in language modeling, where each word of a sentence is represented by an embedding vector [41, 21], we next tested whether adapting pre-trained language models to our upstream data would yield meaningful performance gains, when compared to training models from scratch. To this end, we adapted pre-trained language model variants of GPT-2 [41] and BERT [21] (with 12 hidden layers and an embedding dimension of 768; as provided by Hugging Face's Transformer library [42]) to our upstream dataset (see Appendix B.3). The upstream validation performance of these two language models did not meaningfully improve upon the performance of our models trained from scratch (see section 3.3), at final upstream validation losses of 0.73 (CSM with GPT-2) and 0.80 (Sequence-BERT with BERT) (Appendix Fig. B3).

### 3.5 Downstream adaptation: CSM outperforms other frameworks

Lastly, in our fourth experiment, we evaluated the downstream adaptation performance of our pre-trained models in two benchmark mental state decoding datasets (HCP and MDTB; see section 3.1.2). To gauge the effectiveness of pre-training, we adapted the pre-trained models to varying sizes of the downstream datasets by randomly assigning individuals to distinct training, validation, and test

datasets. Specifically, for each evaluated dataset size, we first randomly selected 10 (HCP) and 3 (MDTB) individuals whose data we use for validation, and 20 (HCP) and 9 (MDTB) other individuals whose data we use for testing, before randomly sampling the given number of individuals whose data we use for training (1 to 48 (HCP) and 11 (MDTB)) from the remaining pool of individuals. We then adapted each pre-trained model with varying learning rates and numbers of training steps to the training data (see Appendix B.4) and used the model variant that achieved the highest final mental state decoding accuracy in the validation data for any further analyses (for an overview of validation decoding accuracies, see Appendix Fig. B6 and B7). For comparison, we also trained a linear baseline model (inline with the state-of-the-art in mental state decoding [43]) and a variant of our decoding head $p(\cdot)$ that we applied directly to the parcellated BOLD sequences $X$ (see Appendix B.5).

Table 1: HCP test decoding performances. Models are trained on varying training dataset sizes ($N = 1, 3, 6, 12, 24, 48$ individuals) and tested on a distinct dataset ($N_{test} = 20$). F1-scores are macro-averaged and reported with standard errors. Bold font indicates best metric value. Asterisks indicate meaningfully better predictive accuracy than the respective second-best model in a McNemar test at $\alpha = 0.00083$ (multiple-comparison corrected: $0.005/6$).

| Framework | Metrics | $N=1$ | $N=3$ | $N=6$ | $N=12$ | $N=24$ | $N=48$ |
|---|---|---|---|---|---|---|---|
| Linear | Acc, F1 | $24.2(\pm.69), 19.1$ | $29.0(\pm.72), 27.1$ | $34.2(\pm.75), 31.8$ | $46.3(\pm.79), 47.0$ | $51.8(\pm.79), 54.8$ | $51.9(\pm.79), 55.6$ |
| $p(X)$ | Acc, F1 | $53.5(\pm.79), \mathbf{59.0}$ | $54.1(\pm.79), 59.5$ | $53.9(\pm.79), 59.4$ | $56.0(\pm.78), 60.8$ | $34.9(\pm.75), 39.9$ | $50.2(\pm.79), 56.5$ |
| Autoencoding | Acc, F1 | $60.2(\pm.77), 47.7$ | $69.3(\pm.73), 60.6$ | $75.6(\pm.68), 67.4$ | $82.9(\pm.60), 77.0$ | $88.1(\pm.51), 84.1$ | $91.5(\pm.44), 88.1$ |
| CSM | Acc, F1 | $\mathbf{64.1(\pm.76)}*, 51.4$ | $\mathbf{81.2(\pm.62)}*, \mathbf{72.9}$ | $\mathbf{86.8(\pm.54)}*, \mathbf{81.0}$ | $\mathbf{90.1(\pm.47)}*, \mathbf{85.8}$ | $\mathbf{92.7(\pm.41)}*, \mathbf{89.2}$ | $\mathbf{94.8(\pm.35)}*, \mathbf{92.0}$ |
| Seq-BERT | Acc, F1 | $39.5(\pm.77), 16.0$ | $37.2(\pm.76), 21.6$ | $52.1(\pm.79), 38.7$ | $73.6(\pm.70), 65.6$ | $73.4(\pm.70), 65.9$ | $89.2(\pm.49), 84.5$ |
| Net-BERT | Acc, F1 | $36.9(\pm.76), 11.9$ | $45.8(\pm.79), 31.4$ | $59.0(\pm.78), 43.5$ | $73.4(\pm.70), 64.4$ | $82.8(\pm.60), 76.5$ | $89.8(\pm.48), 85.2$ |

Table 2: MDTB test decoding performances. Conventions as in Table 1 ($N_{test} = 9$).

| Framework | Metrics | $N=1$ | $N=3$ | $N=6$ | $N=11$ |
|---|---|---|---|---|---|
| Linear | Acc, F1 | $56.1(\pm.47), 42.2$ | $70.2(\pm.44), 65.6$ | $75.6(\pm.41), 72.9$ | $78.1(\pm.39), 77.7$ |
| $p(X)$ | Acc, F1 | $71.8(\pm.43), 65.4$ | $74.5(\pm.42), 70.0$ | $77.6(\pm.40), 76.7$ | $68.0(\pm.44), 59.8$ |
| Autoencoding | Acc, F1 | $68.8(\pm.44), 55.8$ | $78.4(\pm.39), 70.8$ | $83.4(\pm.35), 77.8$ | $86.7(\pm.32), 83.4$ |
| CSM | Acc, F1 | $\mathbf{77.1(\pm.40)}*, \mathbf{69.9}$ | $\mathbf{85.1(\pm.34)}*, \mathbf{83.1}$ | $\mathbf{88.5(\pm.30)}*, \mathbf{87.1}$ | $\mathbf{90.0(\pm.29)}*, \mathbf{89.7}$ |
| Seq-BERT | Acc, F1 | $63.1(\pm.46), 36.9$ | $75.6(\pm.41), 57.9$ | $82.5(\pm.36), 72.5$ | $86.3(\pm.33), 79.6$ |
| Net-BERT | Acc, F1 | $71.6(\pm.43), 54.7$ | $81.1(\pm.37), 72.1$ | $84.6(\pm.34), 78.8$ | $87.5(\pm.32), 84.3$ |

The pre-trained models clearly outperformed the linear baseline model across all training dataset sizes, while their performances also scaled well with the number of individuals in the training data (Tables 1 and 2; at chance-levels of $5.0\%$ (HCP) and $3.9\%$ (MDTB)). Overall, the transformer decoder model trained in the CSM framework clearly outperformed the other models by achieving the highest test decoding accuracies throughout all evaluated sizes of the training dataset. The other pre-trained models performed on par with the decoding head $p(\cdot)$ in smaller training dataset sizes ($\leq 6$ (HCP) and $\leq 3$ (MDTB) individuals), while clearly outperforming it in larger training datasets. For the HCP data, for which other reported decoding performances exist in the literature, the pre-trained models performed on par with models trained on much larger datasets (often comprising data of many hundred individuals [10, 7, 6, 44]), which generally report test decoding accuracies between $80\%$ and $93\%$.

A feature ablation analysis of the models' accurate mental state decoding decisions in both datasets further revealed that these decisions strongly depend on the activity of the occipital and inferior temporal cortex as well as parts of the pre- and postcentral gyrus (see Appendix B.6).

We also tested whether the reported downstream model performances are stable over the non-deterministic aspects of their training [45] by replicating our adaptation analysis with different random seeds and splits of the data (see Appendix B.7). This replication confirmed our results, yielding final test decoding accuracies that were not meaningfully different from our initial analysis (see Appendix Tables A3, A4, A5, A6).

## 4   Conclusion

In this work, we propose a set of novel self-supervised learning frameworks for neuroimaging data that are inspired by prominent learning frameworks in NLP. By the use of these frameworks, we

are able to pre-train DL models on broad, public neuroimaging data, comprising many individuals, experimental domains, and acquisition sites. The pre-trained models generalize well to benchmark mental state decoding datasets and generally outperform baseline models trained from scratch, while models pre-trained in a causal sequence modeling framework clearly outperform the others. We hope that this work will inspire others to explore the benefits and limitations of pre-training in functional neuroimaging at scale, using techniques from self-supervised learning.

**Limitations.** This work neglects a key aspect of mental state decoding, namely, the ability to draw inferences about the association between the decoded mental states and input brain activity from the trained models. First empirical evidence indicates that attribution methods from explainable artificial intelligence (XAI [46]) research are well-suited to provide insights in the mental state decoding decisions of DL models [19, 47]. Further, this work does not provide any insights into how the proposed self-supervised learning frameworks compare to contrastive learning techniques, which have demonstrated great empirical success in computer vision [48] and medical imaging [49].

**Potential negative social impact.** We are currently not aware of any direct negative social impacts that could follow from pre-training DL models on public neuroimaging data that are de-identified and collected and shared with human consent in a way approved by institutional review boards.

**Acknowledgments.** We gratefully acknowledge the support of NIH under No. U54EB020405 (Mobilize), NSF under Nos. 1760950, CCF1763315 (Beyond Sparsity), CCF1563078 (Volume to Velocity), and 1937301 (RTML); ARL under No. W911NF-21-2-0251 (Interactive Human-AI Teaming); ONR under No. N000141712266 (Unifying Weak Supervision); ONR N00014-20-1-2480: Understanding and Applying Non-Euclidean Geometry in Machine Learning; N000142012275 (NEPTUNE); NXP, Xilinx, LETI-CEA, Intel, IBM, Microsoft, NEC, Toshiba, TSMC, ARM, Hitachi, BASF, Accenture, Ericsson, Qualcomm, Analog Devices, Google Cloud, Salesforce, Total, the HAI-GCP Cloud Credits for Research program, the Stanford Data Science Initiative (SDSI), the Texas Advanced Computing Center (TACC) at The University of Texas at Austin, and members of the Stanford DAWN project: Facebook, Google, and VMWare. The OpenNeuro data archive is supported by NIH grant R24MH117179. The U.S. Government is authorized to reproduce and distribute reprints for Governmental purposes notwithstanding any copyright notation thereon. Any opinions, findings, and conclusions or recommendations expressed in this material are those of the authors and do not necessarily reflect the views, policies, or endorsements, either expressed or implied, of NIH, ONR, or the U.S. Government.

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
