# A Data details

## A.1 Upstream datasets

Appendix Table A1 provides an overview of the datasets that we included in our upstream training [50, 51, 52, 53, 54, 55, 34, 56, 57, 37, 58, 59, 60, 61, 62, 63, 38, 35, 36, 64, 65, 66, 67, 68, 69, 70, 71, 72, 73]. The unprocessed fMRI data of all datasets are publicly available (under a Creative Commons CC0 license) through OpenNeuro.org [3] under the specified identifier (ID) of each dataset. All fMRI data were de-identified and collected and shared with human consent in a manner approved by institutional review boards. We did not use any personally identifiable data.

Table A1: Overview of upstream datasets. For each dataset, the OpenNeuro.org identifier and DOI are given as well as the number of individuals and fMRI runs included in our upstream dataset, a brief text descriptor, and the DOI of an associated publication.

| ID | DOI | #Individuals | #Runs | Text descriptor | DOI publication |
|---|---|---|---|---|---|
| ds000003 | 10.18112/openneuro.ds000003.v1.0.0 | 13 | 13 | Rhyme judgment | 10.1162/jocn.2007.19.10.1643 |
| ds000009 | 10.18112/openneuro.ds000009.v1.0.0 | 24 | 144 | The generality of self-control | unpublished |
| ds000030 | 10.18112/openneuro.ds000030.v1.0.0 | 144 | 1029 | UCLA Consortium for Neuropsychiatric Phenomics LA5c Study | 10.1038/sdata.2016.110 |
| ds000113 | 10.18112/openneuro.ds000113.v1.3.0 | 20 | 976 | Study Forrest | 10.1038/sdata.2014.3 |
| ds000140 | 10.18112/openneuro.ds000140.v1.0.0 | 33 | 297 | Distinct brain systems mediate the effects of nociceptive input and self-regulation on pain | 10.1371/journal.pbio.1002036 |
| ds000157 | 10.18112/openneuro.ds000157.v1.0.0 | 28 | 28 | Block design food and nonfood picture viewing task | 10.1016/j.bbr.2013.03.041 |
| ds000212 | 10.18112/openneuro.ds000212.v1.0.0 | 39 | 370 | Moral judgments of intentional and accidental moral violations across Harm and Purity domains | 10.1073/pnas.1207992110 |
| ds000224 | 10.18112/openneuro.ds000224.v1.0.3 | 10 | 767 | The Midnight Scan Club (MSC) dataset | 10.1016/j.neuron.2017.07.011 |
| ds001132 | 10.18112/openneuro.ds001132.v1.0.0 | 15 | 45 | Watching BBC's Sherlock | 10.1038/nn.4450 |
| ds001145 | 10.18112/openneuro.ds001145.v1.0.0 | 24 | 24 | Watching The Twilight Zone | 10.1093/cercor/bhv155 |
| ds001499 | 10.18112/openneuro.ds001499.v1.3.1 | 4 | 515 | BOLD5000 | 10.1038/s41597-019-0052-3 |
| ds001612 | 10.18112/openneuro.ds001612.v1.0.2 | 23 | 135 | Offline replay supports planning in human reinforcement learning | 10.7554/eLife.32548 |
| ds001715 | 10.18112/openneuro.ds001715.v1.0.0 | 34 | 407 | Dissociable neural mechanisms track evidence accumulation for selection of attention versus action | 10.1038/s41467-018-04841-1 |
| ds001734 | 10.18112/openneuro.ds001734.v1.0.5 | 108 | 431 | Neuroimaging Analysis Replication and Prediction Study (NARPS) | 10.1038/s41597-019-0113-7 |
| ds001882 | 10.18112/openneuro.ds001882.v1.0.0 | 19 | 150 | Social Decision-Making Intertemporal Choice Task Dataset | 10.7554/eLife.44939 |
| ds001883 | 10.18112/openneuro.ds001883.v1.0.3 | 20 | 158 | Social Decision-Making Risky Choice Task Dataset | 10.7554/eLife.44939 |
| ds001921 | 10.18112/openneuro.ds001921.v1.0.0 | 15 | 30 | Anterior cingulate engagement in a foraging context reflects choice difficulty, not foraging value (1) | 10.1038/nn.3771 |
| ds001923 | 10.18112/openneuro.ds001923.v1.0.0 | 14 | 42 | Anterior cingulate engagement in a foraging context reflects choice difficulty, not foraging value (2) | 10.1038/nn.3771 |
| ds002306 | 10.18112/openneuro.ds002306.v1.0.3 | 6 | 102 | Over 100 Task fMRI Dataset | 10.1038/s41467-020-14913-w |
| ds002345 | 10.18112/openneuro.ds002345.v1.1.4 | 343 | 861 | Narratives Collection | 10.1038/s41597-021-01033-3 |
| ds002685 | 10.18112/openneuro.ds002685.v1.3.1 | 11 | 1263 | Individual Brain Charting | 10.1038/sdata.2018.105 |
| ds002785 | 10.18112/openneuro.ds002785.v2.0.0 | 216 | 1235 | Amsterdam Open MRI Collection-PIOP1 | 10.1038/s41597-021-00870-6 |
| ds002790 | 10.18112/openneuro.ds002790.v2.0.0 | 225 | 887 | Amsterdam Open MRI Collection-PIOP2 | 10.1038/s41597-021-00870-6 |
| ds002841 | 10.18112/openneuro.ds002841.v1.0.1 | 29 | 169 | Intuitive physics with fMRI | 10.7554/eLife.46619 |
| ds002995 | 10.18112/openneuro.ds002995.v1.0.1 | 18 | 192 | Taste Quality Representation in the Human Brain | 10.1523/JNEUROSCI.1751-19.2019 |
| ds003085 | 10.18112/openneuro.ds003085.v1.0.0 | 39 | 156 | Temporal Dynamics of Emotional Music | 10.1016/j.neuroimage.2019.116512 |
| ds003089 | 10.18112/openneuro.ds003089.v1.0.1 | 20 | 40 | Somatosensory phase-encoded bilateral full-body light touch stimulation | 10.1016/j.neuroimage.2020.117257 |
| ds003148 | 10.18112/openneuro.ds003148.v1.0.1 | 35 | 412 | Neuroimaging evidence for network sampling theory of human intelligence | 10.1038/s41467-021-22199-9 |
| ds003242 | 10.18112/openneuro.ds003242.v1.0.0 | 95 | 598 | MRI data of 40 adult participants in response to a cue induced craving task following food fasting, social isolation and baseline (within-subject design) | 10.1038/s41593-020-00742-z |
| ds003338 | 10.18112/openneuro.ds003338.v1.1.0 | 19 | 116 | Behavioral, physiological, and neural signatures of surprise during naturalistic sports viewing | 10.1016/j.neuron.2020.10.029 |
| ds003340 | 10.18112/openneuro.ds003340.v1.0.2 | 18 | 142 | Tasting Pictures: Viewing Images of Foods Evokes Taste-Quality-Specific Activity in Gustatory Insular Cortex | 10.1073/pnas.2010932118 |
| ds003342 | 10.18112/openneuro.ds003342.v1.0.0 | 18 | 187 | Hand-selective visual regions represent how to grasp 3D tools for use: brain decoding during real actions | 10.1523/JNEUROSCI.0083-21.2021 |
| ds003521 | 10.18112/openneuro.ds003521.v1.0.0 | 35 | 35 | Watching Friday Night Lights (Study 2) | 10.1126/sciadv.abf7129 |
| ds003524 | 10.18112/openneuro.ds003524.v1.0.0 | 12 | 24 | Watching Friday Night Lights (Study 1) | 10.1126/sciadv.abf7129 |

## A.2 Downstream mental states overview

We provide a brief overview of the mental states included in both downstream datasets below. For any further details on the experimental procedures of the datasets, we refer the reader to the original publications [33] (HCP) and [39] (MDTB).

**HCP.** Appendix Table A2 provides an overview of the mental states of each HCP experiment task.

Table A2: HCP mental states. For each task, the mental states and total number of mental states are listed.

| Task | Mental states | Count |
|---|---|---|
| Working memory | body, faces, places, tools | 4 |
| Gambling | win, loss | 2 |
| Motor | left / right finger, left / right toe, tongue | 5 |
| Language | story, math | 2 |
| Social | interaction, no interaction | 2 |
| Relational | relational, matching | 2 |
| Emotion | fear, neutral | 2 |

**MDTB.** The MDTB dataset includes the following set of tasks, each representing one mental state in our analyses (as labeled by the original authors): CPRO, GoNoGo, ToM, actionObservation, affective, arithmetic, checkerBoard, emotionProcess, emotional, intervalTiming, landscapeMovie, mentalRotation, motorImagery, motorSequence, nBack, nBackPic, natureMovie, prediction, rest, respAlt, romanceMovie, spatialMap, spatialNavigation, stroop, verbGeneration, and visualSearch.

## A.3 FMRI data preprocessing

We preprocessed all fMRI data with fMRIPrep (versions 20.2.0 and 20.2.3; a minimal, automated preprocessing pipeline for fMRI data [5]) using fMRIPrep's default settings without FreeSurfer [74] surface preprocessing. We then applied a sequence of additional minimal processing steps to fMRIPrep's derivatives, which included i) spatial smoothing of the fMRI sequences with a 3mm full-width at half maximum Gaussian Kernel, ii) detrending and high-pass filtering (at 0.008 s) of the individual voxel activity time courses, and iii) basic confound removal by regressing out noise in the data related to head movement (as indicated by the six basic motion regressors $x$, $y$, $z$, roll, pitch, and yaw) as well as the mean global signal and mean signal for white matter and cerebrospinal fluid masks (as estimated by fMRIPrep). Lastly, we parcellated each preprocessed fMRI run with the DiFuMo atlas (see section 2.2 of the main text) and standardized the resulting individual network time courses to have a mean of $0$ and unit variance.

# B Experiment details

## B.1 Hyper-parameter evaluation run for largest Sequence-BERT model

The largest model variant of the transformer encoder model that we trained in the Sequence-BERT framework during our hyper-parameter evaluation (see section 3.2 of the main text), with 12 hidden layers and an embedding dimension of 768, did not meaningfully learn over the course of its training. For better visibility of the other model performances, we decided to not include the model's training run in Fig. 3 and are instead showing it in Appendix Fig. B1.

## B.2 BOLD reconstruction error in validation data

To evaluate the BOLD reconstruction performance of our pre-trained models (see section 3.3 of the main text) for different parts of the brain, we computed each models' mean reconstruction error ($L_{rec}$) in the upstream validation data for each network of our BOLD parcellation (see section 2.1 of the main text) and projected the average network reconstruction errors back to the voxel-level (as described in section 2.1 of the main text).

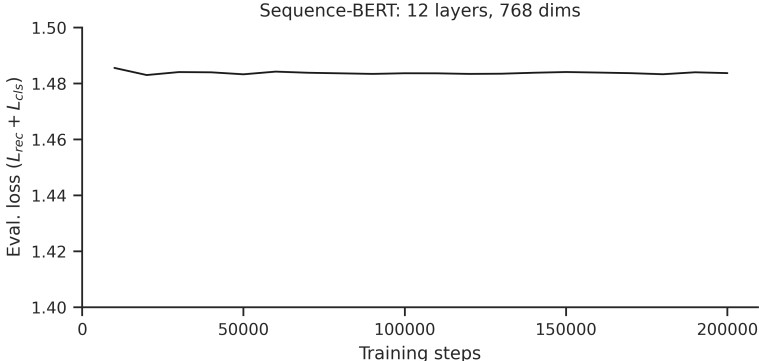

Figure B1: Upstream validation loss of the largest Sequence-BERT model variant with 12 hidden layers and an embedding dimension of 768.

The four pre-trained models exhibit similar distributions of reconstruction error throughout the brain (Appendix Fig. B2), with relatively higher errors in the posterior parietal, occipital, and cingulate cortices as well parts of the limbic system.

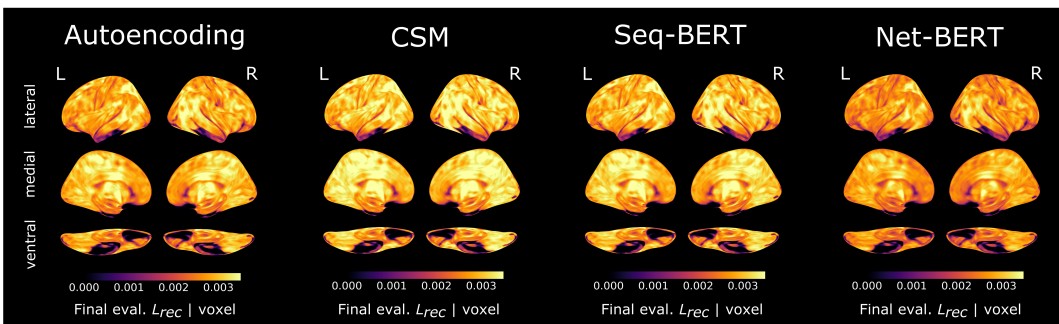

Figure B2: Mean voxel-wise reconstruction error ($L_{rec}$) of the final pre-trained models. Errors are projected onto the inflated cortical surface of the FsAverage template [74].

## B.3   Adaptation of pre-trained language models

We adapted pre-trained language model variants of GPT-2 [41] and BERT [21] (as provided by Hugging Face's Transformer library [42]) to our upstream data in two training phases, using the CSM and Sequence-BERT frameworks, respectively (Appendix Fig. B3): First, we froze all parameters of the two language models and trained them for $20,000$ training steps at a mini-batch size of $512$ and a learning rate of $1e^{-4}$, bringing all other parameters of the CSM and Sequence-BERT frameworks into sensible ranges. After this warmup phase, we continued training both models for a total of $150,000$ training steps of the same mini-batch size, using learning rates of $5e^{-4}$ (GPT-2) and $5e^{-5}$ (BERT) respectively, while allowing all model parameters to change freely.

## B.4   Downstream adaptation of pre-trained models

For each downstream application of our pre-trained models, we evaluated two learning rates ($1e^{-5}$ and $5e^{-5}$) and different training lengths. Specifically, we scaled the number of training steps $N_{step}$ according to the number of subjects $N_{sub}$ in the training data, such that: $N_{step} = 1000 + \eta N_{sub}$, using two $\eta$-values (150 and 300) (see Appendix Fig. B6 and B7). We only report test decoding accuracies for the model variant achieving the highest validation decoding accuracy in this 4-point grid search in the main text (see section 3.5 of the main text).

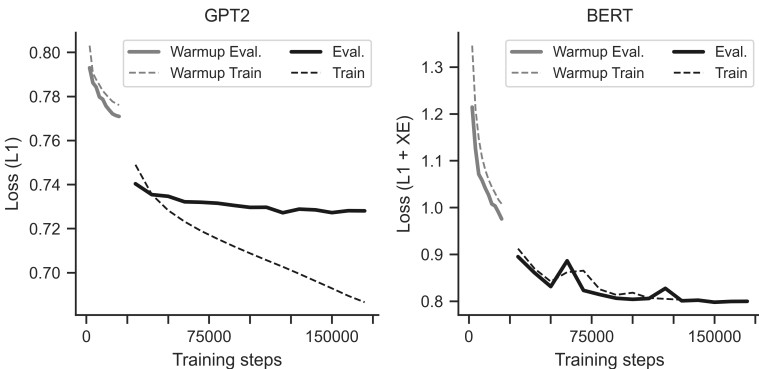

Figure B3: Adapting pre-trained language models to our upstream data in the CSM (GPT-2 [41]) and Sequence-BERT (BERT [21]) frameworks does not yield meaningful performance gains.

### B.5 Baseline models

**Linear baseline.** The linear baseline model forms a decoding decision in two steps: it first aggregates the time course of each network $n \in X$: $a_n = b_n + \sum_t X_{t,n} w_{t,n}$ (with $w$ and $b$ indicating weights and biases) and then predicts a probability $p(a)_i$ that $X$ belongs to mental state $i$ from the distribution of aggregated time courses $a \in \mathbb{R}^n$: $p(a)_i = \sigma(b_i + \sum_n a_n w_{n,i})$, where $\sigma$ represents the $softmax$ function and $w$ and $b$ indicate a second set of weights and biases.

**Decoding head $p(\cdot)$.** We also evaluated the performance of our decoding head $p(\cdot)$ (see section 2.2.1 of the main text) when applied directly to the parcellated BOLD data $X \in \mathbb{R}^{t \times n}$. To allow for the application of $p(\cdot)$ to inputs of varying length, we first averaged the signal of each network $n$ over its time course ($\overline{X}_n = \frac{1}{t} \sum_{i=1}^t X_{i,n}$) before forwarding the time-averaged signal $\overline{X}$ to the decoding head to make a decoding decision for each mental state $i$: $p(\overline{X})_i$.

**Training.** Similar to the other frameworks, we trained both baseline models to minimize a standard cross-entropy loss with additional $L2$-regularization: $L_{cls} = -\sum_i y_i \log p_i + \lambda \sum_i w_i^2$, where $y_i$ indicates a binary variable that equals 1 if $i$ is the correct mental state and 0 otherwise, while $\lambda$ scales the $L2$-regularization strength.

**Hyper-parameter evaluation.** For each application of the baseline models, we evaluated two learning rates ($1e^{-3}$ and $1e^{-4}$), three $L2$-regularisation strengths (0.1, 1, and 10), and two training lengths (1000 and 3000 training steps at a mini-batch size of 512; Note that the baseline models generally required fewer training steps to converge than the pre-trained models; see Appendix Fig. B6 and B7) in a 12-point grid search. As for the pre-trained models, we only report the test decoding accuracy of the model variant achieving the highest validation accuracy in the main text (see section 3.5 of the main text for details on the data split and Appendix Fig. B6 and B7 for an overview of model performances).

### B.6 Downstream feature ablation analysis

To understand which parts of the brain were most relevant for the mental decoding decisions of the adapted models, we performed a feature ablation analysis of the correct test mental state decoding decisions of the best-performing models (that were adapted to the largest training dataset size of both downstream datasets; see Tables 1 and 2 of the main text). Specifically, for each sample of the test datasets, we replaced the signal of individual networks with random Gaussian noise from a standard normal distribution and measured the effect on the model's decoding prediction for the sample's mental state label. Note that we evaluated the predicted logits prior to their $softmax$ scaling. This analysis revealed that the decoding decisions of all pre-trained models are strongly dependent on the signal of the occipital and inferior temporal cortex as well as parts of the pre- and postcentral gyrus in both datasets (see Appendix Fig. B4 and B5).

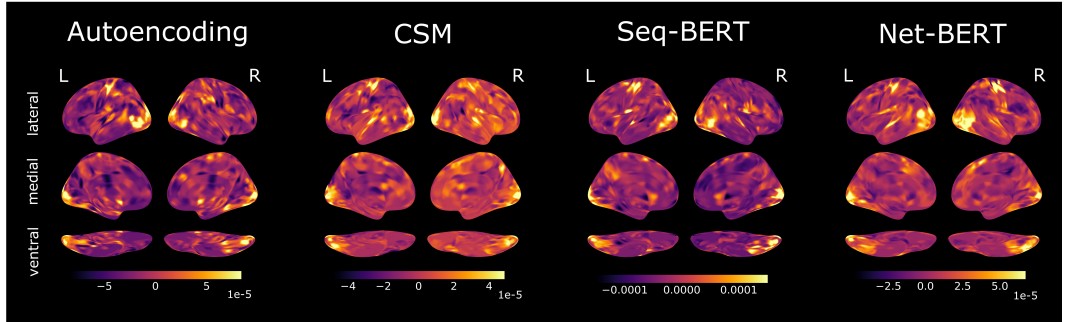

Figure B4: Feature ablation analysis of the pre-trained models for the HCP test dataset. For each sample, we ablate the time course of individual networks and measure the resulting effect on the models' prediction for the mental state associated with the sample. Average changes in output are projected onto the inflated cortical surface of the FsAverage template [74].

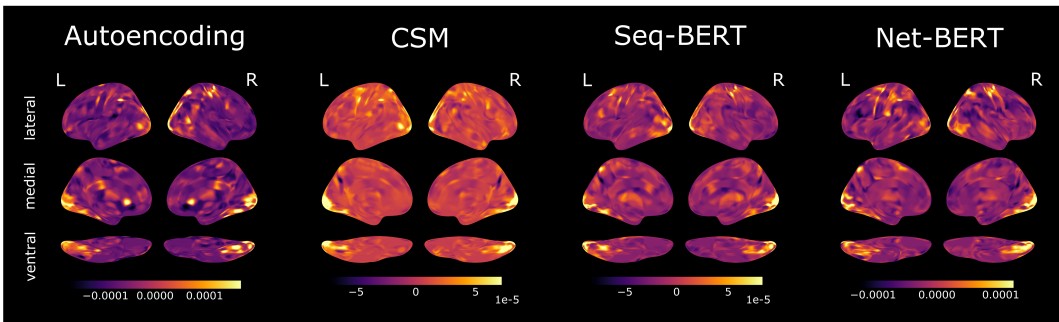

Figure B5: Feature ablation analysis of the pre-trained models for the MDTB test dataset. Conventions as in Appendix Fig. B4.

## B.7 Replication of downstream adaptation analysis

To test for the stability of the reported model performances over the non-deterministic aspects of their training (such as different random weight initializations or random shufflings of the data during training), we replicated our downstream adaptation analysis (see section 3.5 of the main text) with different random splits of the data and different random seeds per split. This replication confirmed our initial results (compare Tables 1 and 2 of the main text with Appendix Tables A3 and A5).

We also tested whether the final test decoding accuracy of each model training run was meaningfully different between our initial analysis (see Appendix Fig. B6 and B7) and the replication (see Appendix Fig. B8 and B9) by computing the difference in final test decoding accuracy between each initial training run and its replication and testing the resulting distribution of test decoding accuracy differences against a mean of 0 in a two-sided t-test (Appendix Tables A4 and A6). The models' final test decoding accuracies were not meaningfully different between our initial analysis and the replication, indicating strong stability of the models' performance over the various non-deterministic aspects of their training.

Table A3: HCP test performances in replication analysis. Conventions as in Table 1.

| Framework | Metrics | $N=1$ | $N=3$ | $N=6$ | $N=12$ | $N=24$ | $N=48$ |
|---|---|---|---|---|---|---|---|
| Linear | Acc, F1 | 33.1($\pm$.74), 28.8 | 36.8($\pm$.76), 35.3 | 44.7($\pm$.79), 45.7 | 49.1($\pm$.79), 50.2 | 50.0($\pm$.79), 50.7 | 55.6($\pm$.79), 53.7 |
| $p(X)$ | Acc, F1 | 53.0($\pm$.79), 58.9 | 50.1($\pm$.79), 58.8 | 53.8($\pm$.78), 61.2 | 55.8($\pm$.79), 61.7 | 35.2($\pm$.76), 40.1 | 51.5($\pm$.79), 59.0 |
| Autoencoding | Acc, F1 | 59.1($\pm$.78), 44.9 | 67.2($\pm$.74), 55.7 | 72.4($\pm$.71), 64.8 | 81.5($\pm$.61), 76.6 | 85.8($\pm$.55), 80.8 | 89.5($\pm$.48), 84.5 |
| CSM | Acc, F1 | **73.4($\pm$.70)**$*$, **61.6** | **78.2($\pm$.65)**$*$, **69.9** | **85.9($\pm$.55)**$*$, **80.9** | **90.6($\pm$.46)**$*$, **86.1** | **91.2($\pm$.45)**$*$, **87.7** | **94.4($\pm$.36)**$*$, **91.6** |
| Seq-BERT | Acc, F1 | 37.1($\pm$.76), 17.3 | 43.3($\pm$.78), 16.1 | 52.2($\pm$.79), 37.8 | 66.9($\pm$.74), 56.2 | 81.2($\pm$.62), 73.5 | 88.6($\pm$.50), 84.1 |
| Net-BERT | Acc, F1 | 36.6($\pm$.76), 13.4 | 49.4($\pm$.79), 27.4 | 57.9($\pm$.78), 46.6 | 68.9($\pm$.73), 61.3 | 77.9($\pm$.66), 70.0 | 87.1($\pm$.53), 81.4 |

Table A4: Statistical comparison of HCP test decoding performances between the initial analysis and replication. Two-sided t-tests compare the the distribution of differences in final test decoding accuracy between each fitting run of the initial analysis and its replication against 0.

| | $N=1$ | $N=3$ | $N=6$ | $N=12$ | $N=24$ | $N=48$ |
|---|---|---|---|---|---|---|
| Linear | $t(11)=-1.34, p=0.21$ | $t(11)=-1.54, p=0.15$ | $t(11)=-4.12, p=0.002$ | $t(11)=-2.88, p=0.02$ | $t(11)=-4.22, p=0.001$ | $t(11)=-3.58, p=0.004$ |
| $p(X)$ | $t(11)=1.08, p=0.30$ | $t(11)=0.73, p=0.48$ | $t(11)=-0.36, p=0.73$ | $t(11)=1.35, p=0.21$ | $t(11)=0.33, p=0.75$ | $t(11)=-1.16, p=0.27$ |
| Autoencoding | $t(3)=6.67, p=0.007$ | $t(3)=0.92, p=0.43$ | $t(3)=0.78, p=0.49$ | $t(3)=1.75, p=0.18$ | $t(3)=4.39, p=0.02$ | $t(3)=0.72, p=0.52$ |
| CSM | $t(3)=-0.86, p=0.45$ | $t(3)=4.32, p=0.02$ | $t(3)=1.55, p=0.22$ | $t(3)=1.28, p=0.29$ | $t(3)=3.67, p=0.03$ | $t(3)=1.55, p=0.22$ |
| Seq-BERT | $t(3)=-0.18, p=0.87$ | $t(3)=0.1, p=0.93$ | $t(3)=-1.56, p=0.22$ | $t(3)=-0.72, p=0.53$ | $t(3)=-3.17, p=0.05$ | $t(3)=-1.28, p=0.29$ |
| Net-BERT | $t(3)=0.19, p=0.86$ | $t(3)=1.28, p=0.29$ | $t(3)=-0.41, p=0.71$ | $t(3)=0.44, p=0.69$ | $t(3)=0.9, p=0.44$ | $t(3)=0.23, p=0.83$ |

Table A5: MDTB test performances in replication analysis. Conventions as in Table 1.

| Framework | Metrics | $N=1$ | $N=3$ | $N=6$ | $N=11$ |
|---|---|---|---|---|---|
| Linear | Acc, F1 | 59.2($\pm$.47), 45.7 | 69.6($\pm$.44), 63.3 | 73.5($\pm$.42), 71.0 | 79.3($\pm$.39), 77.7 |
| $p(X)$ | Acc, F1 | 71.6($\pm$.43), **65.6** | 70.9($\pm$.43), 67.6 | 77.2($\pm$.40), 76.9 | 63.2($\pm$.46), 58.3 |
| Autoencoding | Acc, F1 | 67.9($\pm$.44), 50.8 | 80.9($\pm$.37), 72.2 | 83.8($\pm$.35), 77.3 | 85.2($\pm$.34), 80.4 |
| CSM | Acc, F1 | **73.5($\pm$.42)**, 60.3 | **84.3($\pm$.35)**$*$, **83.5** | **87.2($\pm$.32)**$*$, **85.6** | **90.0($\pm$.27)**$*$, **89.7** |
| Seq-BERT | Acc, F1 | 63.9($\pm$.46), 35.6 | 83.0($\pm$.32), 81.2 | 85.8($\pm$.31), 82.2 | 88.8($\pm$.30), 86.9 |
| Net-BERT | Acc, F1 | 72.0($\pm$.42), 58.6 | 82.8($\pm$.36), 75.0 | 86.0($\pm$.34), 79.5 | 85.2($\pm$.34), 78.4 |

Table A6: Statistical comparison of MDTB test decoding performances between the initial analysis and replication. Conventions as in Table A4.

| | $N=1$ | $N=3$ | $N=6$ | $N=11$ |
|---|---|---|---|---|
| Linear | $t(11)=0.81, p=0.44$ | $t(11)=0.95, p=0.36$ | $t(11)=1.09, p=0.30$ | $t(11)=-0.98, p=0.35$ |
| $p(X)$ | $t(11)=-2.55, p=0.03$ | $t(11)=-0.3, p=0.77$ | $t(11)=0.87, p=0.40$ | $t(11)=-2.58, p=0.03$ |
| Autoencoding | $t(3)=2.1, p=0.13$ | $t(3)=0.35, p=0.75$ | $t(3)=0.15, p=0.89$ | $t(3)=0.47, p=0.67$ |
| CSM | $t(3)=-1.22, p=0.31$ | $t(3)=-1.29, p=0.29$ | $t(3)=-3.29, p=0.05$ | $t(3)=-1.61, p=0.21$ |
| Seq-BERT | $t(3)=-2.0, p=0.14$ | $t(3)=-0.95, p=0.41$ | $t(3)=-0.56, p=0.62$ | $t(3)=-1.32, p=0.28$ |
| Net-BERT | $t(3)=-3.75, p=0.03$ | $t(3)=-4.02, p=0.028$ | $t(3)=-4.62, p=0.02$ | $t(3)=-2.09, p=0.13$ |

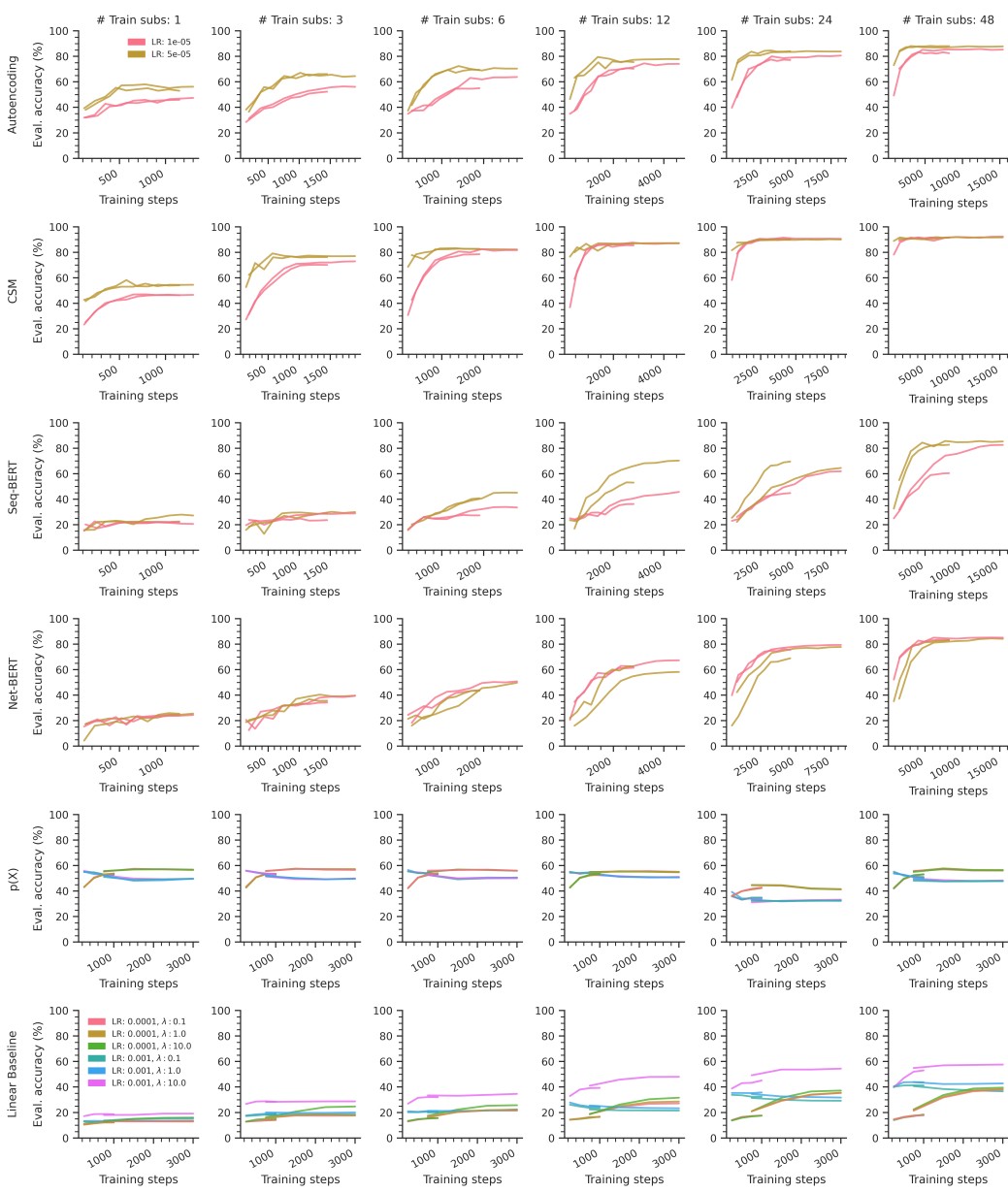

Figure B6: Model validation decoding accuracies during downstream adaptation to HCP data.

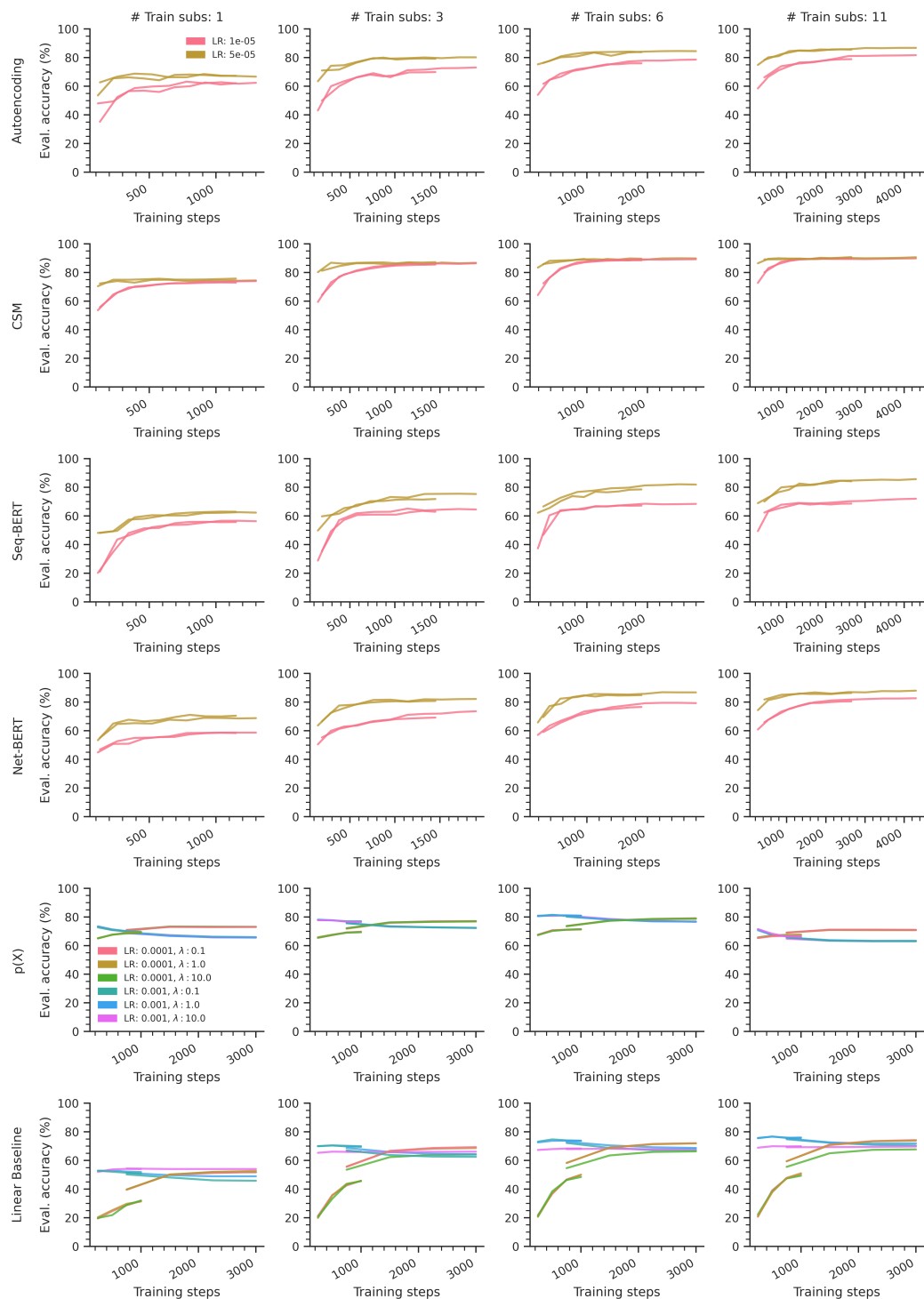

Figure B7: Model validation decoding accuracies during downstream adaptation to MDTB data.

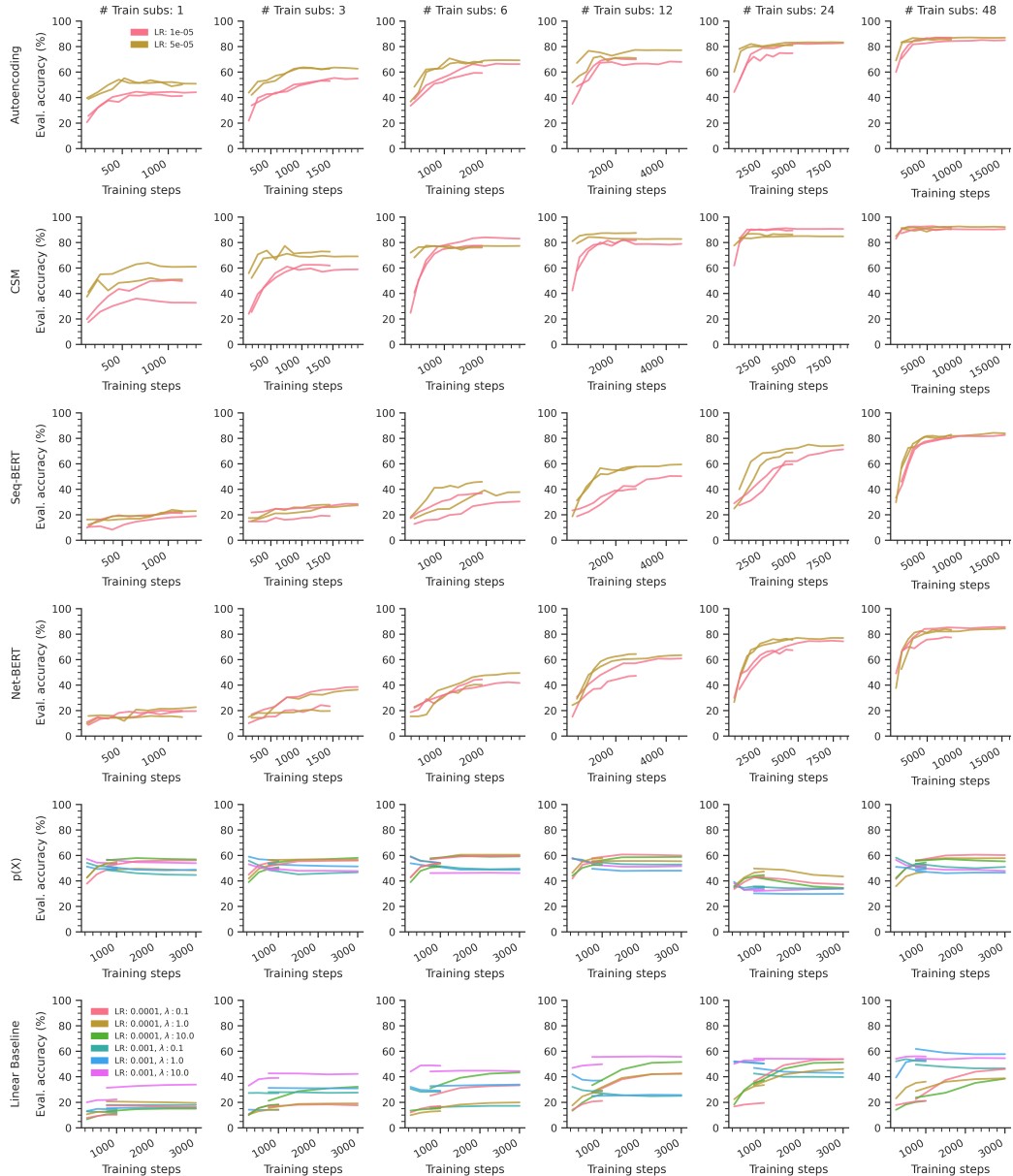

Figure B8: Replication of Appendix Fig. B6 with a different set of random seeds.

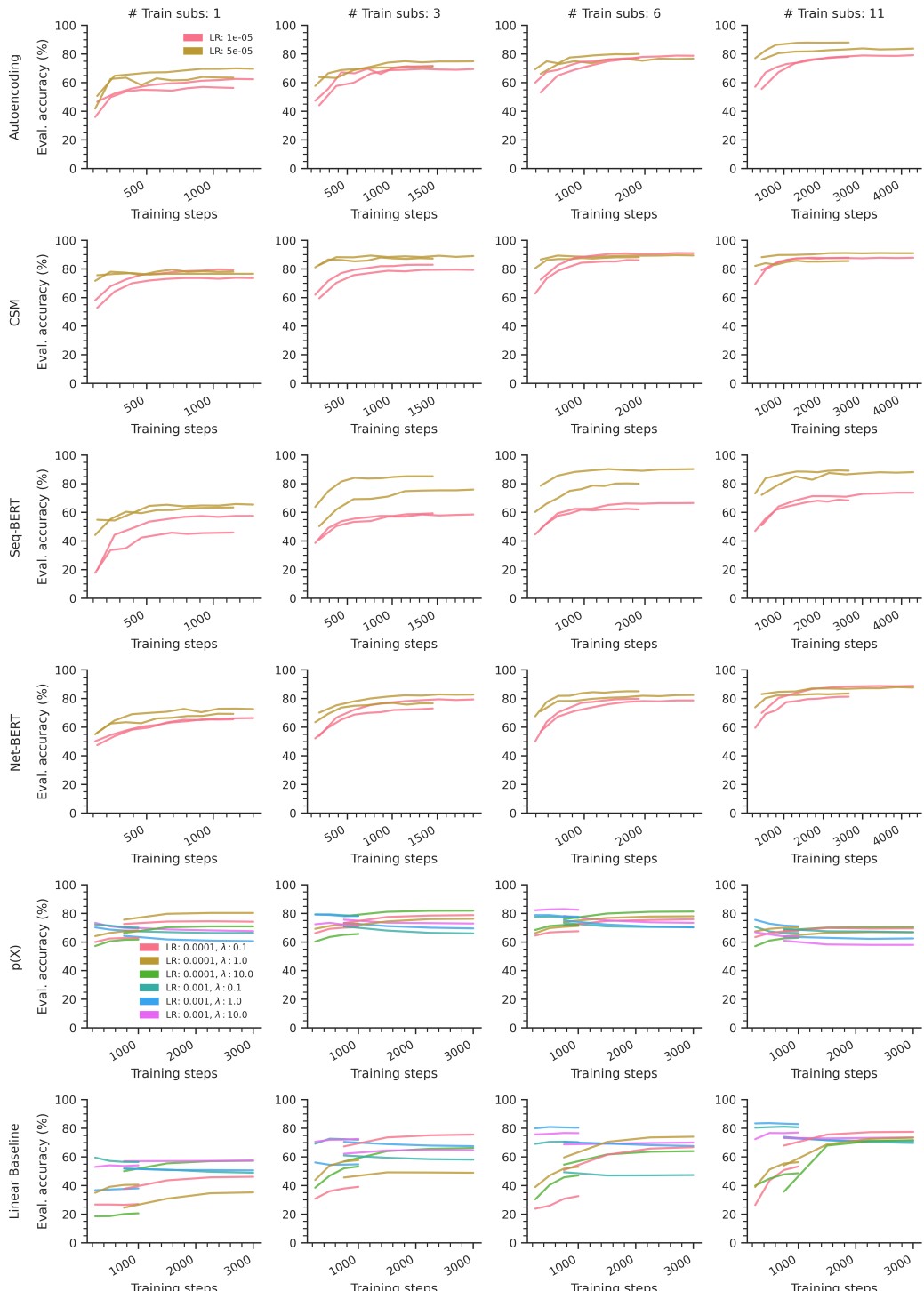

Figure B9: Replication of Appendix Fig. B7 with a different set of random seeds.