# OpenReview forum: "Self-Supervised Learning of Brain Dynamics from Broad Neuroimaging Data"
_NeurIPS.cc/2022/Conference — NeurIPS 2022 Accept_

### Official Review · Reviewer_ca3n · 2022-07-11

**Rating:** 6
**Confidence:** 5
**Soundness:** 3 good
**Presentation:** 3 good
**Contribution:** 3 good

**Summary:**

The authors have extensively experimented with adapting some well-known NLP models to neuroimaging datasets. They have successfully trained autoencoding, causal, and BERT-styled models on either predicting the entire input sequence or predicting parts of the input sequence (mask)
They achieve excellent results in the downstream tasks, especially with the causal seq models, and have shown the generalizability of the trained model to achieve good results on the as-yet unseen datasets.
These types of research are important to further the field of machine learning in neuroimaging as labels are really hard to come by.

**Questions:**

1. Can you explain why you needed a preliminary dimensionality reduction using DiFuMo? And instead just start with X in R^(txp) space where p is the number of voxels (at least in the grey matter)?

2. Why did you restrict yourself to brain decoding and not mental disorders?
3. Why isn't the training code being made available?

**Limitations:**

Brain dynamics often is defined by the interaction between networks. It would be have been great if there were some ablation studies done to check which interaction or single regions were most informative to the downstream task.


**Strengths And Weaknesses:**

Strengths:
A large and extensive dataset was used.
Number of key models from NLP has been adapted.
The pretext task is straight-forward and simple.

Weaknesses:
The pretext tasks don't have anything particular to tie them to predicting brain states. The authors could have used many a datasets out there to predict something useful as in autism or depression as a downstream task.
Why did causal models have to perform better than the rest? No investigation on this has been attempted.

---

> ### Author Response · Authors · 2022-08-02
> **Responses to Questions**
>
> We thank the reviewer for their thoughtful questions, comments, and suggestions. We believe that by addressing these, we could greatly improve the overall quality of our manuscript.
>
> In the following, we will address the reviewer’s questions before addressing other specified weaknesses and limitations in later comments:
>
> **Q1: Can you explain why you needed a preliminary dimensionality reduction using DiFuMo? And instead just start with X in R^(txp) space where p is the number of voxels (at least in the grey matter)?**
>
> **R:** We thank the reviewer for this great question on why we decided to apply the DiFuMo parcellation to our inputs instead of directly using the BOLD sequences $X$ as input. On the  practical side, we decided to use parcellated inputs, as it drastically reduces the number of parameters that we need to learn for our input embedding mechanisms. FMRIPrep’s derivatives, which we use in our analyses, are normalized to the standard “MNI152NLin2009cAsym”-space (at a 2mm resolution). In this space, the gray matter of a volume spans $>200,000$ voxels. Accordingly, our linear embedding mechanism would require $>150,000,000$ parameters to project the inputs to the currently used embedding dimension of $768$ ($>200,000 \times 768$). On the theoretical side, we chose the DiFuMo parcellation because it is very well-suited for our analyses, as it was learned from millions of fMRI volumes across 25 task-based and two resting state fMRI studies that are available on OpenNeuro and which were preprocessed with fMRIPrep (exactly as in our analyses; see [1]). In addition, the original authors have shown that DiFuMo performs extremely well in mental state decoding analyses, when compared to other common choices for brain parcellations (see Fig. 4 in [1]).
>
> [1]. Dadi, K. et al. (2020). Fine-grain atlases of functional modes for fMRI analysis. NeuroImage, 221, 117126.
>
>
> **Q2: Why did you restrict yourself to brain decoding and not mental disorders?**
>
> **R:** We thank the reviewer for this very insightful question on why we decided to focus our efforts on mental state decoding instead of the decoding of mental disorders. We have decided to focus on mental state decoding, as we are concerned about possible negative social consequences that could follow from developing and publicly releasing DL models that are able to decode medical information (such as the presence of or predisposition for mental disorders) from public neuroimaging data. Recent empirical work has already indicated that it is, in some cases, possible to identify individuals from de-identified neuroimaging data (e.g., [1]). DL models that are tailored to the decoding of medical information could thus improve the ability of ill-intended actors to derive medical information about individuals that have been identified from public neuroimaging data.
>
> [1] VanRullen, R., & Reddy, L. (2019). Reconstructing faces from fMRI patterns using deep generative neural networks. Communications biology, 2(1), 1-10.
>
>
> **Q3: Why isn't the training code being made available?**
>
> **R:** We thank the reviewer for raising this question. In line with the NeurIPS guidelines ([https://nips.cc/Conferences/2022/PaperInformation/NeurIPS-FAQ](https://nips.cc/Conferences/2022/PaperInformation/NeurIPS-FAQ)), our code is currently available in the following anonymous repository: [https://osf.io/6xz28/?view\_only=d0536ac81d554a18a9328998d3f9e5c7](https://osf.io/6xz28/?view\_only=d0536ac81d554a18a9328998d3f9e5c7)
>
> Once a review decision has been made, we will also publicly share the code, models, and data on GitHub.

---

> > ### Author Response · Authors · 2022-08-02
> > **Responses to weaknesses and limitations (1/2)**
> >
> > In the following, we will address all weaknesses and limitations stated by the reviewer.
> >
> > **The pretext tasks don't have anything particular to tie them to predicting brain states.**
> >
> > **R:** We thank the reviewer for pointing out that our proposed pre-training tasks are not directly tied to the prediction of mental states. One of the main reasons why we developed two adaptations of BERT for neuroimaging data was that they involve a decoding task (i.e., decoding whether two input sequences belong to the same fMRI run; see section 2.2.4 of our manuscript). Even though this decoding task does not directly involve mental states, it still forces models to learn representations of the data that allow them to make decoding decisions (here, deciding whether two input sequences are related). We would further like to emphasize that supervised pre-training, which would directly link the pre-training to the decoding of mental states, is not easily implemented across multiple functional neuroimaging datasets, for the following reasons:
> >
> > In spite of several attempts [e.g., 1, 2], functional neuroimaging research has yet to adopt standardized definitions of mental states. Without such standardization, it is often unclear whether two experiments from two laboratories elicit the same or different sets of mental states. Imagine two experiments: In one, participants read aloud a sequence of sentences and are then asked to repeat the last word of each sentence. In the other, participants hear a sequence of letters and digits and are then asked to repeat the letters and digits in alphabetical/numerical order. Both datasets might assign the label “working memory” to the underlying mental state. Yet, one can argue that the experiments in fact trigger distinct mental states, as one solely requires temporarily storing information while the other also requires actively manipulating this information (for a more detailed discussion, see [3]).
> >
> > For these reasons, we were interested in developing learning tasks that do not require researchers to assign/derive mental state labels for the training data. To make this more clear to the reader, we now specifically highlight this challenge in the introduction of our revised manuscript: “While supervised training can be fruitful within individual neuroimaging datasets, it is difficult to extend to many datasets, as neuroimaging researchers often do not use standardized labeling schemes for mental states when assigning labels to the mental states of their experiments. This lack of standardization makes it challenging to determine whether two datasets contain the same or distinct sets of mental states (for a detailed discussion of this issue, see 18, 19).” (p. 2, ll. 43-48)
> >
> > [1] Turner, J.A. and Laird, A.R. (2012) The cognitive paradigm ontology: design and application. Neuroinformatics 10, 57–66
> >
> > [2] Poldrack, R.A. et al. (2011) The Cognitive Atlas: Toward a Knowledge Foundation for Cognitive Neuroscience. Front. Neuroinformatics 5,
> >
> > [3] Thomas, A. W., Ré, C., & Poldrack, R. A. (2021). Challenges for cognitive decoding using deep learning methods. arXiv preprint arXiv:2108.06896.
> >
> >
> > **The authors could have used many a datasets out there to predict something useful as in autism or depression as a downstream task.**
> >
> > **R:** We thank the reviewer for suggesting to evaluate our models on clinical use cases, such as the decoding of autism or depression. We provide an extensive answer to this suggestion in our response to Q2 of this reviewer.
> >
> >
> > **Why did causal models have to perform better than the rest? No investigation on this has been attempted.**
> >
> > **R:** We thank the reviewer for raising this great question on why models pre-trained in the causal-sequence-modeling (CSM) framework outperform the others. Our findings are in line with many related empirical findings in natural language processing and computer vision, which show that causal learning frameworks often perform better at scale, when compared to bi-directional masked learning or sequence-to-sequence autoencoding. Compared to the other frameworks, CSM comes closest to training generative models of brain activity, as it trains models to predict the next time point from a given sequence of brain activity. We hypothesize that this generative perspective on the data enables models to better learn the temporal and spatial dynamics of brain activity, when compared to, for example, the masking of individual time points within a sequence, which can often be correctly reconstructed by simply interpolating the signal of neighboring time points. While we currently do not have any empirical evidence for this hypothesis, besides the reported model performances, we are eager to explore it in future work.

---

> > > ### Author Response · Authors · 2022-08-02
> > > **Responses to weaknesses and limitations (2/2)**
> > >
> > > **Brain dynamics often is defined by the interaction between networks. It would be have been great if there were some ablation studies done to check which interaction or single regions were most informative to the downstream task.**
> > >
> > > **R:** We thank the reviewer for this great suggestion to perform ablation studies to gain insights into the models' learned mappings between brain activity and mental states. In line with this suggestion, we have added an ablation study to our revised manuscript (see Appendix A.10). This analysis shows that the decoding decisions of the pre-trained models strongly depend on the signal of the occipital and inferior temporal cortex as well as parts of the pre- and postcentral gyrus in both downstream datasets.

---

### Official Review · Reviewer_JeMm · 2022-07-11

**Rating:** 5
**Confidence:** 5
**Soundness:** 2 fair
**Presentation:** 3 good
**Contribution:** 3 good

**Summary:**

In this work, the authors benchmark a number of NLP architectures in the task of learning representations from fMRI data. The architectures include a seq2seq, transformer, and transformer-based BERT models. Unlike the original NLP models trained with cross entropy loss, the reconstruction loss is used in the presend work instead. The proposed objectives were pre-trained on 34 and evaluated on two publicly available datasets. The results show applicability of self-supervised pre-training for downstream tasks such as mental state decoding. Potentially interesting work and research direction but with a weak evaluation and inconclusive utility of the approach.


**Questions:**

- Would the results and ranking still hold if the embeddings are properly evaluated using metrics not sensitive to a possible data imbalance (see details above)?
- How would the results compare with supervised embeddings (see details above)?
- How does $p(\cdot)$, the same model as used now for everything but the baseline, applied to the input $X$ rank against other models (see details above)?

**Limitations:**

- An inconclusive demonstration of utility of the proposed approach (major).
- Focus on the pre-training approaches based on masking and lack of mention or comparison with contrastive self-supervised learning, which shows strong results in a similar context.

**Strengths And Weaknesses:**

## Strengths
- A novel application of causal and masked data modeling to fMRI data.
- A demonstration of viability of the masked self-supervised training for pre-training of models embedding fMRI, as shown in the transfer learning on a state-decoding task on fMRI. However, the experiments are limited to one downstream task on two datasets. Evaluation and the training is also problematic (see the Weaknesses below)
- A wide variety of considered NLP approaches.

## Weaknesses (approximately in the order of importance)
- Unfair evaluation. Unlike the cross entropy loss in the original NLP applications of the employed models, the reconstruction error introduces a problem in all models but the seq2seq autoencoder, where the bottleneck representation $h$ is used for downstream task. The downstream evaluating model $p(\cdot)$ needs to be a part of the evaluation on the input X, not just the "Linear Baseline" from the paper. Main concern is that the learned embeddings and the input data X are only a single linear transform apart from each other. If not that transform, the CSM and BERT models would be encouraged to learn identity functions, up to a linear autoencoder.
- Inconclusive evaluation. Only accuracy numbers are shown, but the numbers highly dependent on the balance in the dataset. To avoid gross misinterpretations the field uses now standard approaches such as ROC AUC or F1, which in the multiclass case will OVO MACRO or similar. Balanced accuracy is also a better choice. A confusion matrix would reveal some details, but still require a sumarization.
- Evaluation on the downstream task should not simply refer to Appendix but present confidence intervals and statistical significance of improvements. The standard way of comparing performance. Since this is an application paper the rigor of comparison is the most important metric of paper quality. For example, loss curves are unimportant as uninformative, since we're evaluating the utility of considered self-supervised approaches and a closer focus on downstream task, including the model variability.
- Lacking comparison with models pre-trained in supervised manner. Given the data, all of the models (architectures) considered in the manuscript could have been pre-trained with the downstream model attached, i.e. $p(f(X))$ trained to predict the states, but only the obtained embedding used further in evaluation. It may happen that supervised embedding would be as or more efficient than the proposed approach.
- The experiments do not support the claim that training on the homogenous dataset limits the generalizability of the pre-trained models. Moreover, the self-supervised or supervised learning objectives are agnostic to the homogeneity of the dataset. Thus this is a limitation of the experimental setup in the related work rather than the limitations of the pre-training models. Combining multiple inhomogeneous datasets in the medical imaging domain might lead to underspecification and shortcuts that the model can use. Hence, researchers need to control for additional factors (e.g., site effect, age, gender). To support the claim, further experiments are needed to see how the number of datasets and their homogeneity affects the generalizability of self-supervised learning objectives and whether it is a limiting factor.
- The authors claim that the data is minimally-preprocessed. However, this and related work follows = standard preprocessing such as spatial smoothing and detrending. Common choices for preprocessing pipelines are fMRIPrep, C-PAC, or SPM. In addition, the authors use DiFuMo, which is a matrix factorization method, which itself is a way to embed fMRI data. The authors need to clarify what is minimal. Moreover, self-supervised objectives would be agnostic again.
- Note, the statements about the way BERT is trained are not technically correct. BERT is pre-trained either on Masked LM or Next Sentence Prediction. Not Next Sentence Prediction and reconstruction error simultaneously as used in the manuscript.

---

> ### Author Response · Authors · 2022-08-02
> **Responses to Questions (1/2)**
>
> We thank the reviewer for their thoughtful questions, comments, and suggestions. We believe that by addressing these, we could greatly improve the overall quality of our manuscript.
>
> In the following, we will address the reviewer’s questions before addressing other stated weaknesses and limitations in later comments:
>
> **Q1: Would the results and ranking still hold if the embeddings are properly evaluated using metrics not sensitive to a possible data imbalance (see details above)?**
>
> **R:** We thank the reviewer for raising this great question on whether the results of our downstream adaptation analyses hold when using metrics that are not sensitive to a possible imbalance of data labels. In line with this question, we now report F1-scores (macro-averaged) in Tables 1-2 of the revised manuscript as well as standard errors for the decoding accuracies. Importantly, our results hold when taking these additional metrics into account. To further strengthen our analysis, we now also report the results of a statistical comparison of the decoding performance of the best performing model (CSM) in each evaluated training dataset size with the performance of the respective second-best performing model (using the McNemar test). This analysis reveals that the CSM models perform meaningfully better than the respective second-best models across all training dataset sizes (at multiple-comparison corrected alpha levels of 0.00083 (0.005 / 6; HCP) and 0.00125 (0.005/4; MDTB); see Tables 1-2 of the manuscript).
>
>
> **Q2: How would the results compare with supervised embeddings (see details above)?**
>
> **R:** We thank the reviewer for raising this very insightful question on how the reported downstream model performances compare to models pre-trained in a supervised manner (by assigning mental states to each sample of the upstream data and training models to accurately decode these states). Unfortunately, supervised pre-training is not as straightforward across multiple functional neuroimaging datasets as suggested by the reviewer:
>
> In spite of several attempts [e.g., 1, 2], functional neuroimaging research has yet to adopt standardized definitions of mental states. Without such standardization, it is often unclear whether two experiments from two laboratories elicit the same or different sets of mental states. Take the following two experiments as an example: In the first, participants read aloud a sequence of sentences and are then asked to repeat the last word of each sentence. In the second, participants hear a sequence of letters and digits and are then asked to repeat the letters and digits in alphabetical/numerical order. Both datasets might assign the label “working memory” to the mental state, while one can argue that the experiments in fact trigger distinct mental states, as one solely requires temporarily storing information while the other also requires actively manipulating this information (for a more detailed discussion, see [3]).
>
> For these reasons, we believe that self-supervised pre-training is of key importance to functional neuroimaging research, as it enables the pre-training of models at scale across many publicly available fMRI datasets without the need of deriving or applying standardized labeling schemes for mental states. To make this more clear for the reader, we have added the following to the introduction of our revised manuscript: “In addition, researchers often use supervised pre-training techniques by training models to decode specific mental states that are assigned to each data sample (6, 7, 10, 17). While supervised training can be fruitful within individual neuroimaging datasets, it is difficult to extend to many datasets, as neuroimaging researchers often do not use standardized labeling schemes for mental states when assigning labels to the mental states of their experiments. This lack of standardization makes it challenging to determine whether two datasets contain the same or distinct sets of mental states
>  (for a detailed discussion of this issue, see 18, 19).” (p. 2, ll. 42-48)
>
> [1] Turner, J.A. and Laird, A.R. (2012) The cognitive paradigm ontology: design and application. Neuroinformatics 10, 57–66
>
> [2] Poldrack, R.A. et al. (2011) The Cognitive Atlas: Toward a Knowledge Foundation for Cognitive Neuroscience. Front. Neuroinformatics 5,
>
> [3] Thomas, A. W., Ré, C., & Poldrack, R. A. (2021). Challenges for cognitive decoding using deep learning methods. arXiv preprint arXiv:2108.06896.

---

> > ### Author Response · Authors · 2022-08-02
> > **Responses to Questions (2/2)**
> >
> > **Q3: How does p(), the same model as used now for everything but the baseline, applied to the input rank against other models (see details above)?**
> >
> > **R:** We thank the reviewer for raising this great question on how the decoding performance of our decoding head $p(\cdot)$ (see section 2.2.1 of the manuscript) compares to the decoding performance of the pre-trained models when applied directly to the input data $X$. In line with this suggestion, we have trained $p(\cdot)$ directly on the input data and included its test decoding performance in Tables 1-2 of our revised manuscript (for methodological details on the training procedure, see Appendix A8). The pre-trained CSM model clearly  outperforms $p(\cdot)$, while the other pre-trained models perform on par with $p(\cdot)$ in smaller training datasets ($N \leq 6$ (HCP) and $N\leq3$ (MDTB) ) and clearly outperform it in larger datasets.

---

> > > ### Author Response · Authors · 2022-08-02
> > > **Responses to weaknesses and limitations (1/3)**
> > >
> > > In the following, we will address all weaknesses and limitations stated by the reviewer.
> > >
> > > **Unfair evaluation. Unlike the cross entropy loss in the original NLP applications of the employed models, the reconstruction error introduces a problem in all models but the seq2seq autoencoder, where the bottleneck representation h is used for downstream task. The downstream evaluating model p() needs to be a part of the evaluation on the input X, not just the "Linear Baseline" from the paper. Main concern is that the learned embeddings and the input data X are only a single linear transform apart from each other. If not that transform, the CSM and BERT models would be encouraged to learn identity functions, up to a linear autoencoder.**
> > >
> > > **R:** We thank the reviewer for these comments on our choice of a reconstruction error loss for the pre-training tasks. We believe that these comments are based on a misunderstanding of our methodology, which we would like to clarify: The reviewer states that the reconstruction loss might encourage models (with the exception of the autoencoder) to learn identify functions of their input, as the input $X$ and the input’s embedding representation $E^X$ are only a linear transform apart from one another. We would like to make clear that this is not the case, as we train models to reconstruct MASKED time points of the input sequences. Specifically, we replace the time points that are to be reconstructed with a learned mask embedding (akin to the mask embeddings used in language modeling; see Figure 1) and train models to predict the data that was replaced by the mask embedding. The data that is to be reconstructed is therefore not part of the input to the model and could not be reconstructed with an identity function (an identity function would return the mask embedding).
> > >
> > >
> > > **Inconclusive evaluation. Only accuracy numbers are shown, but the numbers highly dependent on the balance in the dataset. To avoid gross misinterpretations the field uses now standard approaches such as ROC AUC or F1, which in the multiclass case will OVO MACRO or similar. Balanced accuracy is also a better choice. A confusion matrix would reveal some details, but still require a sumarization.**
> > >
> > > **R:**  We thank the reviewer for this great suggestion to use accuracy metrics that are sensitive to a possible class imbalance in our downstream evaluation. In line with this suggestion, we now report F1-scores (macro-averaged) in Tables 1-2 of our revised manuscript. Importantly, our results hold when accounting for the F1-scores.
> > >
> > >
> > > **Evaluation on the downstream task should not simply refer to Appendix but present confidence intervals and statistical significance of improvements. The standard way of comparing performance. Since this is an application paper the rigor of comparison is the most important metric of paper quality. For example, loss curves are unimportant as uninformative, since we're evaluating the utility of considered self-supervised approaches and a closer focus on downstream task, including the model variability.**
> > >
> > > **R:** We thank the reviewer for suggesting to present confidence intervals and statistical significance tests for the results of our downstream evaluation. In line with this suggestion, we now report standard errors for the reported decoding accuracies. In addition, we compare the decoding performance of the best performing model (CSM) in each training dataset size to the performance of the respectively second-best performing model by means of the McNemar test. The CSM models perform meaningfully better than the respective second-best models across all training dataset sizes (at multiple-comparison corrected alpha levels of $0.00083$ (0.005 / 6; HCP) and $0.00125$ (0.005/4; MDTB); see Tables 1-2 of the manuscript).
> > >
> > >
> > > **Lacking comparison with models pre-trained in supervised manner. Given the data, all of the models (architectures) considered in the manuscript could have been pre-trained with the downstream model attached, i.e.  p(f(x)) trained to predict the states, but only the obtained embedding used further in evaluation. It may happen that supervised embedding would be as or more efficient than the proposed approach.**
> > >
> > > **R:** We thank the reviewer for suggesting to compare the performance of our pre-trained models to the performance of models that are trained in a supervised manner. We would like to emphasize that supervised pre-training across multiple neuroimaging datasets is not as straightforward as suggested by the reviewer, as it is often unclear whether two experiments use the same or different labels for the same mental state (for a more detailed explanation, see our response to Q 2 of this reviewer). For this reason, we believe that self-supervised pre-training is of key relevance to functional neuroimaging research, as it can be easily applied to the wealth of existing public neuroimaging data.

---

> > > > ### Author Response · Authors · 2022-08-02
> > > > **Responses to weaknesses and limitations (2/3)**
> > > >
> > > > **The experiments do not support the claim that training on the homogenous dataset limits the generalizability of the pre-trained models. Moreover, the self-supervised or supervised learning objectives are agnostic to the homogeneity of the dataset. Thus this is a limitation of the experimental setup in the related work rather than the limitations of the pre-training models. Combining multiple inhomogeneous datasets in the medical imaging domain might lead to underspecification and shortcuts that the model can use. Hence, researchers need to control for additional factors (e.g., site effect, age, gender). To support the claim, further experiments are needed to see how the number of datasets and their homogeneity affects the generalizability of self-supervised learning objectives and whether it is a limiting factor.**
> > > >
> > > > **R:** We thank the reviewer for pointing out that our experiments do not support the claim that pre-training on homogeneous datasets limits the generalizability of the resulting models. While our findings do not directly address this issue, we would like to point out that a wealth of other empirical work has shown that individual neuroimaging datasets often contain systematic biases (e.g., resulting from the imaging site or experimental paradigm), which limit the generalizability of models trained on these datasets [1-5]. We would also like to point out that self-supervised learning is not agnostic to these kinds of systematic biases: For example, if a model is trained in a self-supervised manner on a neuroimaging dataset that contains systematic biases in imaging noise (or other kinds of systematic biases, e.g., resulting from specific eye or hand movements that are linked to the experimental paradigm), the model will incorporate these biases in its learned representations, which limits the generalizability of these representations (and thereby the model’s performance) to datasets that do not include these biases. To make this more clear for the reader, we have re-written this part of our introduction, which now states: “Yet, much of this work has either pre-trained models on large but homogenous datasets, such as data from many individuals who all perform the same few tasks at the same few acquisition sites (10, 8), which can limit the generalizability of the resulting models due to systematic biases that can be contained in homogeneous datasets (e.g., specific to the acquisition site or experimental paradigm; 11, 12, 13, 14 15), or required highly-preprocessed input data (e.g., using statistical maps summarizing the measured sequences of brain activity; 16)” (p. 1-2, ll. 36-42).
> > > >
> > > > [1] Chen, P. H. C. et al. (2015). A reduced-dimension fMRI shared response model. Advances in Neural Information Processing Systems, 28.
> > > >
> > > > [2] Sahoo, D., & Davatzikos, C. (2021). Learning Robust Hierarchical Patterns of Human Brain across Many fMRI Studies. Advances in Neural Information Processing Systems, 34, 29034-29048.
> > > >
> > > > [3] Yousefnezhad, T. M. et al. (2020). Shared space transfer learning for analyzing multi-site fmri data. Advances in Neural Information Processing Systems, 33, 15990-16000.
> > > >
> > > > [4] Yousefnezhad, M. et al. (2020). Supervised Hyperalignment for Multisubject fMRI Data Alignment. IEEE Transactions on Cognitive and Developmental Systems, 13(3), 475-490.
> > > >
> > > > [5] Kragel, P. A. et al. D. (2018). Generalizable representations of pain, cognitive control, and negative emotion in medial frontal cortex. Nature neuroscience, 21(2), 283-289.
> > > >
> > > >
> > > > **The authors claim that the data is minimally-preprocessed. However, this and related work follows = standard preprocessing such as spatial smoothing and detrending. Common choices for preprocessing pipelines are fMRIPrep, C-PAC, or SPM. In addition, the authors use DiFuMo, which is a matrix factorization method, which itself is a way to embed fMRI data. The authors need to clarify what is minimal. Moreover, self-supervised objectives would be agnostic again.**
> > > >
> > > > **R:** We thank the reviewer for pointing out that our parcelated input data can be viewed as not minimally-preprocessed. We agree with the reviewer that this is our subjective interpretation and have therefore removed any statements from our manuscript claiming that our frameworks use minimally-preprocessed fMRI data.

---

> > > > > ### Author Response · Authors · 2022-08-02
> > > > > **Responses to weaknesses and limitations (3/3)**
> > > > >
> > > > > **Note, the statements about the way BERT is trained are not technically correct. BERT is pre-trained either on Masked LM or Next Sentence Prediction. Not Next Sentence Prediction and reconstruction error simultaneously as used in the manuscript.**
> > > > >
> > > > > **R:** We would like to point out that this statement of the reviewer is factually not correct. BERT pre-training simultaneously considers the masked language modeling (MLM) and next sentence prediction (NSP) task, as indicated by BERT’s pre-training loss, which represents the sum of the MLM and NSP losses (see Appendix A.2 of the original BERT paper [1]).
> > > > >
> > > > > [1] Devlin, J. et al. (2018). Bert: Pre-training of deep bidirectional transformers for language understanding. arXiv preprint arXiv:1810.04805.
> > > > >
> > > > >
> > > > > **Focus on the pre-training approaches based on masking and lack of mention or comparison with contrastive self-supervised learning, which shows strong results in a similar context.**
> > > > >
> > > > > **R:**  We thank the reviewer for pointing out that our previous manuscript did not mention contrastive self-supervised learning as an alternative to the proposed masked learning strategies. We have now added a statement to the limitations section of our work (section 4), which states: “Similarly, this work currently does not provide any insights into how the proposed self-supervised learning frameworks compare to contrastive learning techniques, which are often used in computer vision (46) and medical imaging (47)”. (p. 9, ll. 370-372)

---

> > > > > > ### Comment · Reviewer_JeMm · 2022-08-08
> > > > > > **details**
> > > > > >
> > > > > > re: BERT Thanks for pointing out this hidden note in the paper. The main BERT manuscript does not indicate that there is potentially an extra hyperparameter: the proportion in which these losses are mixed.
> > > > > >
> > > > > > Note, reference 46 that you use as a way to point to contrastive SSL explicitly says on page 9:
> > > > > > > We leave exploration of contrastive pre-training (Chen et al., 2020b; He et al., 2020; Bachman et al., 2019; Henaff et al., 2020) to future work.
> > > > > >
> > > > > > I find including this reference misleading in this context.
> > > > > >
> > > > > > Reference 47 used for biomedical contrastive learning mentions "Contrastive" SSL only once in the entire manuscript only as a small entry in Table 6.

---

> > > > > > > ### Author Response · Authors · 2022-08-08
> > > > > > > **Response to reviewer details**
> > > > > > >
> > > > > > > We thank the reviewer for their response and for acknowledging our rebuttal!
> > > > > > >
> > > > > > > **re: BERT loss**: Given that no additional hyper-parameter for the mixing of the two loss components is described in the original BERT paper, we assume that the authors did not use such an additional parameter (as suggested by the reviewer). This assumption is also supported by HuggingFace’s BERT implementation, which simply adds the masked-language-modeling and next-sentence-prediction losses (see line 1139 in [https://github.com/huggingface/transformers/blob/v4.21.1/src/transformers/models/bert/modeling_bert.py#L1053](https://github.com/huggingface/transformers/blob/v4.21.1/src/transformers/models/bert/modeling_bert.py#L1053))
> > > > > > >
> > > > > > > **re: Contrastive pre-training references**: In line with the reviewer’s comments, we have updated the mentioned references to:
> > > > > > >
> > > > > > > 46: Chen, T. et al. (2020). A simple framework for contrastive learning of visual representations. In International conference on machine learning, pp. 1597-1607, PMLR.
> > > > > > >
> > > > > > > 47: Chaitanya, K. et al. (2020). Contrastive learning of global and local features for medical image segmentation with limited annotations. In Advances in Neural Information Processing Systems 33, pp. 12546-12558, Curran Associates.

---

> > > ### Comment · Reviewer_JeMm · 2022-08-08
> > > **Q3**
> > >
> > > Thank you for adding $p(\cdot)$ to the comparison table. It is informative.

---

> > ### Comment · Reviewer_JeMm · 2022-08-08
> > **Q1**
> >
> > Thank you for adding the standard error and the F1 score. Any specific reason not to have standard error on the F1 score? If this is the space in the table, I would recommend dropping accuracy as anyway it is a misleading measure.

---

> > > ### Author Response · Authors · 2022-08-09
> > > **Response to reviewer remark**
> > >
> > > We thank the reviewer for this suggestion.
> > >
> > > Unfortunately, we do not have enough time left in this revision to properly estimate standard errors for the reported F1-scores (e.g., with bootstrapping). To our knowledge, the standard error formula for binomial random variables (such as predictive accuracy) does not generalize to the F1-score (as it represents a non-linear combination of such binomial random variables). Yet, we do believe that the combination of predictive accuracy and SE as well as F1-score provides enough information for the reader to accurately judge the performance of the evaluated models. We would also like to point out that predictive accuracy is a common performance metric in similar studies (e.g., [1]).
> > >
> > > Considering the discussion period deadline, we look forward to receiving your feedback on our response. If you have any other questions about the paper, please also let us know.
> > >
> > >
> > > [1] Mensch, A. (2017). Learning neural representations of human cognition across many fMRI studies. Advances in neural information processing systems, 30.

---

> ### Comment · Reviewer_JeMm · 2022-08-09
> **concluding**
>
> I thank the authors for responding and addressing some of my concerns. I improve my score, but see below.
>
> My major concern---inconclusive evaluation---is still there. It is difficult to assess the utility of the new approach without knowing the supervised bound. The explanation provided for Q2 may either mean that the application chosen to demonstrate the power of the approach is suboptimal as it prevents clear evaluation, or that the chosen task is unsuitable for training generalizable models, as the lack of label harmonization restricts a model within a dataset. The latter makes me question the point of the exercise, even though I agree with the authors that evaluation is not easy. My remaining concerns revolve around evaluation as well.
>
> To summarize, this paper does not provide enough evidence that the neural NLP-type pretraining is generalizable. I am afraid if accepted the paper will be used as a reference for this. Although the authors are careful in their claims, the structure of the paper and experiments will unfortunately lead many to accept the yet unjustified claim.

---

> > ### Author Response · Authors · 2022-08-09
> > **Response to concluding remark**
> >
> > We thank the reviewer for these final remarks and for reconsidering their initial rating of our manuscript.
> >
> > While supervised pre-training is extremely challenging (if not impossible) to implement across many neuroimaging datasets (as outlined in our response to Q2), we would like to point out that other empirical work (e.g., [1]) has evaluated supervised pre-training solely within the Human Connectome Project (HCP) data, which we use in our downstream evaluation. There, the authors pre-train a deep learning model in the decoding of the HCP’s mental states, using a training dataset with >800 individuals, and subsequently evaluate the performance of the pre-trained model in the decoding of these mental states in a test dataset with distinct individuals. Their pre-trained model achieves an average test decoding accuracy of up to ~90%, which is lower than the decoding performance of our best-performing pre-trained models when these are adapted to the data of only 48 individuals (see Table 1; the CSM model even achieves test decoding accuracies of ~90% when adapted to the data of only 12 individuals). Importantly, our pre-training data does not include any data of the HCP.
> >
> > While this comparison across studies does not fully satisfy a comparison of our self-supervised pre-training methodology to a potential supervised alternative, we hope that it provides for the reviewer some evidence in favor of the utility of our approach.
> >
> > [1] Zhang, Y. et al. (2021). Functional annotation of human cognitive states using deep graph convolution. NeuroImage, 231, 117847.

---

### Official Review · Reviewer_R4Ef · 2022-07-14

**Rating:** 8
**Confidence:** 4
**Soundness:** 4 excellent
**Presentation:** 4 excellent
**Contribution:** 4 excellent

**Summary:**

In this paper, different self-supervised learning architectures are adopted from NLP literature to predict brain activity. Specifically,

1. Autoencoders using LSTMS
2. GPT-style autoregressive models
3. BERT-style masked autoregressive model and a next sentence prediction task adapted to fMRI data (are the two sequences from the same dataset or not)
4. Same as 3 but a particular network (akin to an emebdding dimension) is dropped from all input embeddings

The paper explores training all 4 architectures using data from a large collection of fMRI experiments, and finds that all models train well. While there is no clear effect of model depth, models with higher capacity perform better.

Finally, the pretrained models are evaluated on two downstream datasets (for task classification) and are all found to beat traditional linear baselines.

**Questions:**

Questions:
1. (Lines 237-238) Isn’t 1-5 TRs too small a gap considering HRF? Have the authors experimented with longer gaps?

1. Does the model use low-level cues like different repetition times for the task equivalent to next-sentence-prediction? One way to analyze this would be to correlate CE with some measure of difference in the TRs. Similarly, what about cues like participant ID or task type given that specific networks will be active depending on the task?
2. I am not sure I follow the experiment in section 3.4. If the experiment is to use GPT/BERT pretrained on text and then finetuned on the fMRI setup, why would that work?
3. How is the data quality/SNR related to training/validation/test loss? This was not clear to me but did the original pretraining also split train/val/test such that no participant contributed data to more than one set?

Misc. questions unrelated to my views of the current manuscript:
1. What is the relationship of ablating differentiating diff. networks on the downstream task accuracy? Is there a possibility for meaningful relationships between the same?
2. Relatedly, what is the impact of ablating a specific network on the MAD of different networks in the output? Is there some functional, anatomical organization to the same?

**Limitations:**

-NA-

**Strengths And Weaknesses:**

Edit after rebuttal: I believe that the authors have sufficiently answered my questions and thought the response on using pertained GPT representation was especially interesting. I agree with other reviewers that in its current form, the usability of the networks for real-world downstream tasks is limited. However, I believe this paper is, to the best of my knowledge, highly novel and will spur interesting further research. I recommend it for acceptance.

#########################################################################################################

Strengths:
1. Clarity: This paper was very well written, easy to understand and well visualized. I enjoyed reading it!
2. Quality: I found the results presented convincing, although there is a lot to unpack and it is not fully apparent to me how the choice of hyper parameters, data construction schemes etc. affect the observed results
3. Originality & Significance: To the best of my knowledge, this is the first study to explore such pretraining schemes for fMRI data. It is a very exciting and significant direction for future research!

Weaknesses:
1. It took me a while to understand the need for both positional encodings and the TR encodings. Clarifying this earlier would improve readability in my opinion.
2. I would urge the authors to add standard error to the values in fig.5 over many random seeds.
3. Several (all?) of the analyses do not currently report standard error. Given the number of design choices made throughout, I believe this would be an important addition to improve the paper.

---

> ### Author Response · Authors · 2022-08-02
> **Responses to Questions (1/2)**
>
> We thank the reviewer for their thoughtful questions, comments, and suggestions. We believe that by addressing these, we were able to greatly improve the overall quality of our manuscript.
>
> In the following, we will address the reviewer’s questions before addressing the stated weaknesses and limitations in later comments:
>
> **Q1: (Lines 237-238) Isn’t 1-5 TRs too small a gap considering HRF? Have the authors experimented with longer gaps?**
>
> **R:** We thank the reviewer for raising this great question on why we did not leave longer TR gaps between the two sequences that we sample for the training of BERT-style models. The fMRI datasets that we included in our upstream training were collected with repetition times of 1 s to 4 s. Accordingly, a random gap of 1-5 TRs between two sequences corresponds to a time gap of 1-20 s. The HRF typically has a temporal span of up to 15 s. To our understanding, a random gap of 1-5 TRs between two sequences should therefore be long enough to ensure that the two sequences cannot be reliably connected by interpolating the HRF, especially as the HRF’s onset will not always align with the end of the first sequence. We have therefore not experimented with longer gaps.
>
>
> **Q2: Does the model use low-level cues like different repetition times for the task equivalent to next-sentence-prediction? One way to analyze this would be to correlate CE with some measure of difference in the TRs. Similarly, what about cues like participant ID or task type given that specific networks will be active depending on the task?**
>
> **R:** We thank the reviewer for raising these very insightful questions on whether the models use low-level cues, such as TRs, participant ID or task type, for their mental state decoding decisions in our downstream analyses.
>
> We would like to begin by pointing out that neither TRs (as indicated by the TR embeddings) nor participant IDs are informative about the presence of these mental states in our analyses:
>
> *TRs*: All BOLD data of each downstream dataset were collected with the exact same repetition time (HCP: 0.72 s, MDTB: 1 s). In addition, when assigning TR embeddings to the time points of an input sequence, we always relabel these relative to the onset of the input sequence. For example, when sampling an input sequence from a MDTB run file from 10s to 20 s, we assign TR embeddings to this sequence that correspond to time points 0, 1, 2, …, 10 s. The TR embeddings therefore do not carry any information about the position of the sequence in the underlying fMRI run. For these two reasons, the TR embeddings do not contain any information about the mental states that are to be decoded.
>
> *Participant ID*: While it is in theory possible that models learns to identify individuals from the input sequences (even though the participant IDs are not directly given to the models), this information would not be helpful for the decoding of mental states in our  downstream datasets, as all participants of each dataset conducted the same experimental paradigm, resulting in the same distribution of mental states (ie., labels) for each participant.
>
> *Task type*: A wealth of other empirical work has shown that the mental states of our two downstream datasets are associated with different patterns of brain activity (e.g., [1-2]). Given that the pre-trained models perform well in decoding these mental states, we assume that they also correctly associate these states with such patterns of brain activity. Importantly, no information about the task, besides the TR, is provided to the models during training.
>
> [1] Barch, D. M. et al. (2013). Function in the human connectome: task-fMRI and individual differences in behavior. Neuroimage, 80, 169-189.
>
> [2] King, M. et al. (2019). Functional boundaries in the human cerebellum revealed by a multi-domain task battery. Nature neuroscience, 22(8), 1371-1378.

---

> > ### Author Response · Authors · 2022-08-02
> > **Responses to Questions (2/2)**
> >
> > **Q3: I am not sure I follow the experiment in section 3.4. If the experiment is to use GPT/BERT pretrained on text and then finetuned on the fMRI setup, why would that work?**
> >
> > **R:** We thank the reviewer for raising this great question on why language models pretrained on text data should generalize to our BOLD data. On a high level, sentences represent a type of sequence data, similar to how BOLD data represent a type of sequence data. Given that our input embedding has a very similar format as the embedding of words in sentences, by encoding each time point of a sequence as a vector, we were wondering whether models that are pre-trained on language data (such as GPT and BERT) would generalize to our data, simply because they have learned the core capabilities needed to learn from and process sequence data in this format. To make this more clear for the reader, we have extended our description of these experiments in the manuscript, which now states: “Given that the formatting of our input embedding representations $E^X$ (see section 2.2.1) is highly similar to the embedding representations of sentences in language modeling, where each word of a sentence is represented by a vector, we next tested whether adapting pre-trained language models to our upstream data would yield meaningful performance gains, when compared to training models from scratch.” (p. 7-8, ll. 314-318)
> >
> >
> > **Q4: How is the data quality/SNR related to training/validation/test loss? This was not clear to me but did the original pretraining also split train/val/test such that no participant contributed data to more than one set?**
> >
> > **R:** We thank the reviewer for raising these very insightful questions. During pre-training, we separate the upstream data into distinct training and validation datasets by randomly assigning 5% of the upstream data files (each corresponding to one fMRI run) to the validation dataset and assigning all remaining data files to a distinct training dataset. The upstream data is thereby not split by individuals but by experimental runs. Importantly, separation by experimental run is an established data split strategy in neuroimaging research [1].
> >
> > To also provide some insight into how the fMRI data quality relates to the models’ pre-training losses, we performed an exploratory analysis of the fMRI data of the PIOP2 dataset of the Amsterdam Open MRI Collection (AOMIC) (OpenNeuro ID: ds002790), which spans several hundred individuals. Specifically, we correlated the reconstruction loss of the CSM and autoencoding frameworks for 10,000 random samples with the temporal signal-to-noise ratio (tSNR) of the associated fMRI run files (as estimated by MRIQC; [2]), Note that we excluded the BERT-style models from this analysis, as their pre-training involves the random combination of different BOLD sequences (see section 2.2.4 and 2.2.5 of the manuscript). Overall, this analysis revealed a positive association between reconstruction loss and tSNR, with higher reconstruction errors for fMRI files with more temporal noise:
> >
> > CSM: Pearson’s $r(9998) = 0.0388, p = 0.0001$
> >
> > Autoencoding : Pearson’s $r(9998) = 0.2581, p < 0.0001$
> >
> > [1] Varoquaux, G. (2018). Cross-validation failure: Small sample sizes lead to large error bars. Neuroimage, 180, 68-77.
> >
> > [2] Esteban, O. et al. (2017). MRIQC: Advancing the automatic prediction of image quality in MRI from unseen sites. PloS one, 12(9), e0184661.

---

> > > ### Author Response · Authors · 2022-08-02
> > > **Responses to weaknesses and limitations**
> > >
> > > **It took me a while to understand the need for both positional encodings and the TR encodings. Clarifying this earlier would improve readability in my opinion.**
> > >
> > > **R:** We thank the reviewer for pointing out that our previous manuscript was unclear on why TR embeddings are needed over the positional embeddings used by Transformer models. To make this more clear for the reader, we have adapted the wording in our revised manuscript, which now states “Importantly, TR embeddings carry different information than the position embeddings used in Transformer models (28), as the same position in an input can correspond to different time points, depending on the fMRI's sampling frequency.” (p. 3 , ll. 107-109).
> > >
> > >
> > > **I would urge the authors to add standard error to the values in fig.5 over many random seeds.**
> > >
> > > **R:** We thank the reviewer for suggesting to test our downstream adaptation results for non-deterministic training effects (as resulting from varying random seeds). We would like to point out that our downstream adaptation analysis is computationally more costly than it might seem to the reader at first sight, as we run four model adaptation runs for each pre-trained model and 12 fitting runs for each of the two baseline models (we included an additional baseline model based on Q1 of reviewer #2) to determine model performances for each evaluated size of the training datasets. This results in a total of 240 fitting runs for the HCP adaptation analysis and 160 fitting runs for the MDTB adaptation analysis. Given that we are limited in time and computational budget during this revision, we could not replicate each of these runs over many random seeds, as suggested by the reviewer. Instead, we replicated the entire analysis once for each dataset, with different random seeds and splits of the data (see Appendix Tables A3 and A5 of the manuscript), resulting in a replication of each initial fitting run with a different random seed. We then tested whether the final test decoding accuracies of the fitting runs were meaningfully different between the initial analysis and the replication (see Appendix Tables A4 and A6 of the manuscript). This analysis revealed no meaningful differences in final test decoding accuracies for the pre-trained models in all training dataset sizes, indicating that our reported model performances are stable across the non-deterministic aspects of their training.
> > >
> > >
> > > **Several (all?) of the analyses do not currently report standard error. Given the number of design choices made throughout, I believe this would be an important addition to improve the paper.**
> > >
> > > **R:** We thank the reviewer for bringing to our attention that the analyses of our previous manuscript did not indicate standard errors. In line with this suggestion, we now include standard errors for the model decoding accuracies reported in our downstream analyses (see Tables 1-2 of the manuscript). We also report respective F1-scores (macro-averaged) to give the reader more insight into the models’ performances.
> > >
> > > We decided not to include standard errors for the evaluation losses plotted in Figures 3 and 4 of the manuscript for better visibility and because we are not interested in comparing upstream model performances between the learning frameworks (due to different specifications of the loss functions).
> > >
> > >
> > > ### Misc. questions unrelated to my views of the current manuscript:
> > >
> > > **What is the relationship of ablating differentiating diff. networks on the downstream task accuracy? Is there a possibility for meaningful relationships between the same?**
> > >
> > > **R:** We thank the reviewer for raising this thoughtful question. To provide some insight into the contributions of different networks to the model’s decoding performance, we have added an ablation study to our revised manuscript (see Appendix A.10 of our manuscript). This analysis shows that the decoding decisions of the pre-trained models strongly depend on the signal of the occipital and inferior temporal cortex as well as parts of the pre- and postcentral gyrus in both downstream datasets.
> > >
> > >
> > > **Relatedly, what is the impact of ablating a specific network on the MAD of different networks in the output? Is there some functional, anatomical organization to the same?**
> > >
> > > **R:** We thank the reviewer for raising this insightful question on the contribution of individual networks to the models’ upstream reconstruction losses. Unfortunately, we currently do not have any insights into these contributions, but are eager to explore these in future work.

---

### Author Response · Authors · 2022-08-02
**Revision Summary - Thanks to all reviewers!**

We thank the reviewers for the very insightful feedback, comments, and constructive reviews. We have tried to address all of the reviewers’ remarks to the best of our abilities and think that this has helped us greatly improve the overall quality of our manuscript.

Before providing detailed responses to the individual remarks of the reviewers, we would like to briefly summarize the main changes to our manuscript resulting from the reviews:

1. **Improved evaluation:** To provide better insight into the mental state decoding performance of the pre-trained models, we replaced Figure 5 of our previous manuscript with two Tables (Tables 1-2), which now report standard errors for all decoding accuracies, F1-scores, and the results of a statistical comparison of the two highest decoding performances for each evaluated size of the training dataset. Importantly, all of our results hold when taking these additional metrics into account.

2. **Extended evaluation:** We have added a new baseline model to our downstream analysis (section 3.5), namely, the decoding head $p(\cdot)$ that we apply directly to the parcellated BOLD sequences X (for further details, see Appendix A.8).

3. **Ensuring reproducibility:** To test for the reproducibility of our downstream analysis, we replicated it for both datasets with different random seeds and splits of the data (see Appendix A.11) and found no meaningful differences in the final test decoding accuracies between the initial analysis and replication (see Appendix Tables A4 and A6).

4. **Novel insights:** We perform an ablation study of the mental state decoding decisions of the pre-trained models to provide further insight into the parts of the brain that contribute most to their decisions (see Appendix A.10). We find that these are strongly driven by signals of the occipital and inferior temporal cortex as well as parts of the pre- and postcentral gyrus.

5. **Improved clarity throughout:** We have made several edits to the text of our manuscript, based on the reviewers’ remarks, to improve its overall clarity for the reader

We hope that the reviewers will share our enthusiasm for our revised manuscript and are looking forward to continuing a constructive review process.

---

### Meta-Review · Area_Chair_J6cc · 2022-08-27

**Recommendation:** Accept
**Confidence:** Less certain

**Metareview:**

This submission solicited interesting discussions between the reviewers and the authors, and was seen as bringing interesting ideas to the table. The ideas of the masked self-supervised pre-training for brain imaging is an exciting one.

One strong concern is that the evaluation is not very conclusive, and it is not clear that the present work gives solid evidence that the pre-training actually is beneficial.

**Award:**

No

---

### Decision · Program_Chairs · 2022-09-14

Accept